# A modeling framework to understand historical and projected ocean climate change in large coupled ensembles

Yona Silvy[1], Clément Rousset[1], Eric Guilyardi[1,2], Jean-Baptiste Sallée[1], Juliette Mignot[1], Christian Ethé[3], and Gurvan Madec[1]

[1]LOCEAN-IPSL, Laboratoire d'Océanographie et du Climat: Expérimentation et Approches Numériques, Sorbonne Université/CNRS/IRD/MNHN, Paris, France
[2]NCAS-Climate, University of Reading, Reading, UK
[3]IPSL, Institut Pierre-Simon Laplace, Sorbonne Université, CNRS, Paris, France

**Correspondence:** Yona Silvy (yona.silvy@locean.ipsl.fr)

**Abstract.** The ocean responds to climate change through modifications of heat, freshwater and momentum fluxes at its boundaries. Disentangling the specific role of each of these contributors in shaping the changes of the thermohaline structure of the ocean is central for our process-understanding of climate change, and requires the design of specific numerical experiments. While it has been partly addressed by modeling studies using idealized $CO_2$ forcings, the time evolution of these individual contributions during historical and projected climate change is however lacking. Here, we propose a novel modeling framework to isolate these contributions in coupled climate models for which large ensembles of historical and scenario simulations are available. The first step consists in reproducing a coupled pre-industrial control simulation with an ocean-only configuration, forced by prescribed fluxes at its interface, diagnosed from the coupled model. In a second step, we extract the external forcing perturbations from the historical+scenario ensemble of coupled simulations, and we add them to the prescribed fluxes of the ocean-only configuration. We then successfully replicate the ocean's response to historical and projected climate change in the coupled model during 1850-2100. In a third step, this full response is decomposed in sensitivity experiments in which the forcing perturbations are applied individually to the heat, freshwater and momentum fluxes. Passive tracers of temperature and salinity are implemented to discriminate the addition of heat and freshwater flux anomalies from the redistribution of pre-industrial heat and salt content in response to ocean circulation changes. Here, we first present this general framework and then apply it to the IPSL-CM6A-LR model and its ocean component NEMO3.6. This framework brings new opportunities to precisely explore the mechanisms driving historical and projected ocean changes within single climate models.

## 1 Introduction

The thermohaline structure of the ocean has been particularly affected by human-made climate change: observations of the upper ocean since the mid-20[th] century show an unabated warming, and large-scale salinity changes coherent with an intensification of the hydrological cycle (Fox-Kemper et al., 2021). These changes are consistent with the effects of rising greenhouse gas concentrations in the atmosphere, and are projected to amplify in response to continuing emissions (Fox-Kemper et al., 2021). The spatial patterns of ocean temperature and salinity modifications in response to this forcing are important to under-

stand as they have widespread consequences, including regional sea level rise, one of the major risks associated with climate change for human societies and ecosystems living in coastal areas (IPCC, 2022). Importantly, these spatial patterns are set not only via changes in local heat and freshwater fluxes, but also via changes in large-scale ocean circulation, which redistribute the heat and salt internally. However, the combination of physical drivers causing these changes and their spatial distribution is still unclear.

Ocean general circulation models and climate models are both useful tools to investigate physical processes as they allow for different hypotheses to be tested, by, for example, decomposing potential physical drivers. Numerical experiments have thus been designed to explore the role of individual surface fluxes and/or ocean circulation changes in driving regional ocean heat content change in response to rising $CO_2$ atmospheric concentrations (e.g. Mikolajewicz and Voss (2000); Banks and Gregory (2006); Fyfe et al. (2007); Xie and Vallis (2012); Winton et al. (2013); Marshall et al. (2015); Gregory et al. (2016); Armour et al. (2016); Garuba and Klinger (2016, 2018); Liu et al. (2018); Zanna et al. (2019); Todd et al. (2020); Hu et al. (2020)). Other studies have also explored how salinity patterns were driven by a combination of changes in these surface fluxes (Lago et al., 2016; Zika et al., 2018; Shi et al., 2020). A gap however remains regarding the precise attribution of the ocean response to these different forcings during the historical period and future projections. Indeed, most of the aforementioned studies looked at the response to idealized forcings and/or at multi-decadal to centennial scale. In particular, the FAFMIP (Flux-Anomaly-Forced Model Intercomparison Project, Gregory et al. (2016)) protocol proposes a framework to investigate the ocean response to individual perturbations in surface heat, freshwater and wind stress fluxes in global ocean or coupled climate models. The perturbations are applied as a step forcing, and are constructed as a multi-model mean anomaly from a time of doubling atmospheric $CO_2$ concentrations in idealized simulations in which $CO_2$ increases at a rate of 1% per year. One important goal of FAFMIP is thus to investigate the uncertainty in ocean responses across models to a unique set of perturbations. Strictly speaking, these perturbations are thus not coherent with the climate of each individual climate model, and the FAFMIP protocol does not enable to attribute the mechanisms responsible for ocean change within single coupled climate models. Furthermore, the FAFMIP protocol does not focus on the historical period but on an idealized forcing. Finally, the amplitude and balance of the perturbations in the different surface fluxes in response to historical emissions could look quite different at these timescales than their amplitude in response to strong, idealized forcings. It is precisely under transient historical forcings that the Earth system has been seeing unprecedented changes, including in the ocean. Ocean temperature and salinity changes of the past several decades have already become greater than background climate variations, as simulated by multiple climate models (Silvy et al., 2020). Yet, the processes driving these ocean changes at realistic historical timescales, and particularly at the time they emerge from unforced climate variability, are not fully understood.

Here, we propose a novel modeling framework aiming at understanding the leading ocean processes responsible for climate change signals to emerge from climate variability in the ocean, in response to historical+future scenario forcings. Specifically, we aim to delineate how, within the simulated historical and future climate of single coupled climate models, changes in surface heat fluxes, freshwater fluxes and winds affect the patterns and timescales of ocean temperature and salinity changes. One of the issues will be to also track their relation to changes in the large-scale ocean circulation. The targeted time period for our investigation ranges from the pre-industrial period to the end of the projected 21$^{st}$ century, i.e. 1850-2100, under

realistic, i.e. non-idealised, greenhouse gas concentration pathways. For this, we have designed numerical experiments in a stand-alone configuration of an ocean general circulation model, inherited from a parent fully-coupled model. In section 2, we present the general protocol before focusing on a single global climate model. In sections 3, 4, and 5, we describe practically the implementation steps of the experiments, before concluding in section 6. Companion papers are underway to present the results of these simulations (e.g. Silvy et al. (2022)).

## 2 Experimental design and model

### 2.1 General presentation

Our goal here is to investigate and isolate the mechanisms responsible for changes in ocean temperature and salinity and their emergence from internal variability in historical+scenario simulations of single climate models. To do so, we focus on the response of the ocean to the perturbations of the surface fluxes in freshwater, heat and momentum taken separately. The strategy we propose is to reproduce the 1850-2100 ocean response to climate change of a coupled model with a novel ocean-only setup, which allows to then perform a series of sensitivity experiments where perturbations in surface heat, freshwater and wind stress fluxes are used separately as boundary conditions for the ocean. In this subsection, we will present this strategy for a generic AOGCM (Atmosphere Ocean General Circulation Model), before detailing all the steps with the IPSL-CM6A-LR model in the following one.

One important constraint of the simulations we want to perform concerns background internal variability, i.e. unforced variability intrinsic to the climate system. Internal variability can play an important role in determining the timing of emergence of changes in the climate system, as highlighted by the differences within members of the same model historical+scenario ensemble (Lehner et al., 2017; Silvy et al., 2020). Indeed, historical and scenario simulations of coupled climate models are generally performed several times with the exact same external forcings (anthropogenic emissions and variations in natural factors), but initialized at different pre-industrial climate states so that the response of all of these "members" differs only because of their different phase of internal variability. To extract what is referred to as the "externally-forced" response, the common procedure consists in taking the mean across enough members so that these unforced variations are averaged out, i.e. what is left is the response to external forcings uncontaminated by internal variations (Deser et al., 2020; Milinski et al., 2020).

In this study, we want to investigate the ocean response to historical+scenario-like simulations where the externally-forced perturbations of each surface flux are imposed separately. These perturbations are obtained by computing the ensemble mean of each surface flux (heat, freshwater, stress) in the historical+scenario simulations of the AOGCM under consideration. This entails having several members available, which is the case for many models that took part in the 6[th] phase of the Coupled Model Intercomparison Project (CMIP6, Eyring et al. (2016)). Furthermore, we want to cleanly compare the timing of externally-forced ocean changes in-between sensitivity experiments, i.e. without contamination from internal variability that could yield different emergence times unrelated to external forcings. One way to do that would be to run the AOGCM in its pre-industrial control state, with a prescribed perturbation for each surface flux individually, and run several members of each experiment

to extract the externally-forced response. However, this has strong disadvantages. Because the ocean properties would change under such perturbations, feedbacks would modify the perturbations from their original values in the historical+scenario simulations. That would make comparisons between simulations highly complicated. In particular, to prevent the negative feedback on the heat flux perturbation stemming from its interaction with sea surface temperature in a coupled configuration, methods have been proposed that make the implementation of these perturbations not straightforward (Bouttes and Gregory, 2014). Also, running 251 years (1850-2100 period) of an AOGCM, for several members and several experiments is computationally very expensive.

Instead, we propose another approach, with an ocean-only configuration, that allows extracting the forced response with only one experiment per targeted flux perturbation (i.e. only one member). Simulations are performed with the ocean component (OGCM) of the parent AOGCM. The OGCM configuration is the same as for the AOGCM, except that the ocean is forced by prescribed fluxes at its boundaries instead of being coupled to an atmosphere. This is done in the following way (see schematic in Fig. 1):

**Step 1.** Setting up the CTL experiment (with the ocean model only) forced by surface fluxes diagnosed from a 251-year pre-industrial control simulation of the AOGCM (piControl, constant external forcings). The goal for this ocean-only CTL is to inherit the mean climate and internal variability of the coupled model piControl, thereby providing a background climate with the same phases of variability for the sensitivity experiments. The piControl fluxes are thus imposed at high enough frequency so as to reproduce the internal variability as accurately as possible (here we chose 3-hourly, i.e. twice the coupling frequency) at the liquid ocean interface (below the atmosphere and the sea-ice), during 251 years. Since the fluxes are extracted below sea-ice and imposed on the ocean, the sea-ice model component is excluded from the oceanic configuration. Note however that the imposed fluxes are extracted from a fully coupled AOGCM which itself includes an interactive sea-ice module, so that the ocean-only component we run is impacted by both atmosphere and sea-ice fluxes. This CTL experiment is initialized similarly as its parent piControl simulation.

**Step 2.** Setting up the ALL experiment, forced with the exact same fluxes as CTL, plus a perturbation component on each surface flux (heat, freshwater, winds; Fig. 1) constructed from the AOGCM's monthly-mean fluxes during the historical+scenario simulations (1850-2100). These fluxes are averaged over all possible historical+scenario members of the AOGCM so as to extract the best estimate of the externally-forced response, and the anomaly is then constructed relative to 1850-1899. One should use an AOGCM that has performed a large ensemble of historical simulations (1850-2014 in CMIP6), followed in the remaining of the 21$^{st}$ century by a large ensemble of scenario simulations for 2015-2100 (SSPs in CMIP6; Shared Socioeconomic Pathways; Gidden et al. (2019)). The ALL experiment is designed to reproduce the historical+scenario response of the coupled model. It is initialized as the CTL and thus more generally as any historical member from a parent piControl state, with its own initial state.

**Step 3.** Setting up the sensitivity experiments (HEAT, WATER, STRESS). This step is similar to step 2 but the different contributions to the perturbation (heat, freshwater and winds) are imposed separately. We also perform a BUOY experiment which includes both the heat and freshwater fluxes perturbations.

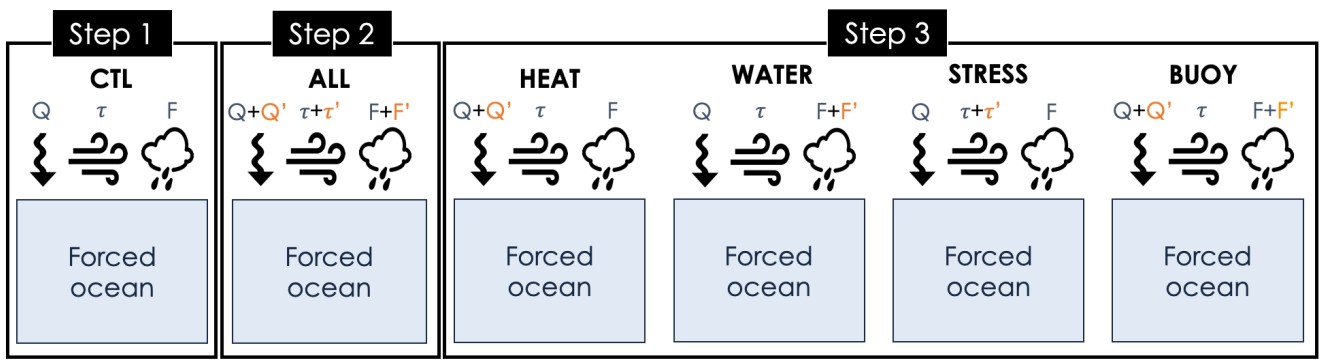

**Figure 1.** Schematic of the ocean-only simulations

All experiments are thus aligned on the internal variability of the 251-year portion of the piControl initially chosen for CTL, with the additional perturbation (externally-forced) components slowly taking effect. This means that theoretically, at any time step, taking the difference of any simulation that includes a perturbed component from CTL isolates the externally-forced signal, since they have concurrent phases of internal variability. The fact that all surface fluxes are prescribed without any feedbacks ensures that the forcings do not differ from what the ocean has seen in the coupled model. Under the hypothesis of

linear additivity between the surface fluxes, the sensitivity experiments allow decomposing the response of the ALL experiment, while keeping identical phases of internal variability. This hypothesis and its limits are addressed in section 4.3.

    Step 1 and Step 2 can be technically validated by ensuring that the response is close to the parent AOGCM simulations, respectively the piControl, and the historical+scenario ensemble mean. With the proposed setup, if the simulations were perfect, the difference ALL-CTL would capture the coupled model's ensemble mean anomaly from the historical+scenario period. In

practice, they are not perfect because of technical compromises such as the forcing frequency. In the application of the protocol which we present in this paper, we will see that the differences remain smaller than the ensemble spread almost everywhere in the ocean. The sensitivity experiments cannot be validated using simulations from the coupled model, but we apply simple checks ensuring that a) heat budget is identical between HEAT, BUOY and ALL, and between WATER, STRESS and CTL and b) freshwater budget is identical between WATER, BUOY and ALL, and between HEAT, STRESS and CTL.

This protocol bears similarities with the ocean-only Flux-Anomaly-Forced Model Intercomparison Project (FAFMIP, Gregory et al. (2016); Todd et al. (2020), www.fafmip.org). However, significant differences exist as already addressed in the Introduction, and which we summarize in the discussion section. We also implemented two passive tracers in each of these experiments to discriminate between added versus redistributed heat and freshwater anomalies, which we describe in section 5.

We now proceed to apply this protocol effectively to the IPSL-CM6A-LR coupled model and the NEMO3.6 ocean model. We describe the implementation in practice and show some validation. Nevertheless, as described above, the experimental design could be applied to any coupled model and its ocean-only configuration, as long as the externally-forced historical and future response can be extracted. The historical+scenario large ensemble approach seems to be the most accurate way to isolate the forced response compared to e.g. fitting a 4th order polynomial to a single member (Lehner et al., 2020).

## 2.2 Application to IPSL-CM6A-LR

We use the IPSL-CM6A-LR coupled model (Boucher et al., 2020) developed by the Institut Pierre-Simon Laplace modeling center for CMIP6. It is composed of the LMDZ6A atmospheric model (Hourdin et al., 2020), the ORCHIDEE land surface model (Krinner et al., 2005) version 2.0 and the NEMO3.6 ocean model (Madec et al., 2017); see schematic in Fig. 2. The atmospheric component has a horizontal resolution of 2.5°x1.3° on a regular latitude-longitude grid and 79 vertical layers, while the ocean component uses the eORCA1 tripolar grid with a nominal horizontal resolution of 1° refined to 1/3° at the equator, with 75 vertical levels with varying thicknesses (1m at surface to 200m at deepest levels). Indeed the vertical layers are time dependent, with a nonlinear evolving free surface using a variable volume formulation. The ocean physics component (OPA) of NEMO3.6 is coupled to the LIM3 sea-ice model (Rousset et al., 2015) and to the PISCES-v2 biogeochemical model (Aumont et al., 2015). The oceanic equation of state is estimated with a polynomial representation of TEOS-10 (Roquet et al., 2015); the model prognostic fields are thus conservative temperature and absolute salinity. The different schemes and parameterizations employed in the eORCA1 configuration and used in IPSL-CM6A-LR are described in Boucher et al. (2020) and additional details can be found in Madec et al. (2017). We also use the ocean physics component in stand-alone mode, without the sea-ice nor the biogeochemistry, with the same configuration eORCA1 as in the coupled model.

This work is performed in the context of the simulated climate of the IPSL-CM6A-LR coupled model. For the purpose of the CMIP6 exercise, after a long spin-up, multiple experiments were already conducted with this coupled model (Boucher et al., 2020) including: 2000 years of piControl simulation, 32 members of the historical period (1850-2014) that was extended to 2059 under the ssp245 (Shared Socioeconomic Pathway 2-4.5, Gidden et al. (2019)) scenario, and projections (2015-2100) with only 11 members. Thus, to apply the framework described previously, we have a large ensemble of 32 members over the period 1850-2059 to construct the surface flux perturbations. Since this is the period over which a large ensemble exists, we will mainly focus on this one. Because the regular ssp245 scenario was performed to 2100, but with 11 members only, we use the ensemble mean of these 11 members over 2060-2100 to complete our simulations to 2100. Using 11 members after 2060 is certainly less accurate than 32 members. Indeed, we have shown in previous work that the envelope of interannual variability of the piControl was well sampled with 30 members, and in particular that the piControl interannual standard deviation of ocean temperature and salinity is well reproduced by the intermember standard deviation (Silvy, 2022).

Because the piControl of the IPSL-CM6A-LR published for CMIP6 did not have all the needed outputs at high-enough frequency, we also had to run before-hand another piControl simulation with the IPSL-CM6A-LR coupled model to save the fluxes at the liquid ocean interface at 3-hourly frequency as well as other components (Fig. 2). This piControl is initialized

from the same spin-up simulation used as restart state in the piControl r1i1p1f1 published for CMIP6, but ran on a different supercomputer. Our piControl ran for 401 years, but only the 3-hourly fluxes from the last 251 years are used in the forced experiments presented here. The IPSL-CM6A-LR model has a systematic quasi-linear drift in global-mean ocean temperature (equivalent to -0.13 W.m$^{-2}$, Mignot et al. (2021)), passed on to this piControl and subsequently to the CTL and other ocean-only experiments, due to a negative incoming heat flux (see heat budget in Table 2). Unless noted otherwise, this drift is not removed in the figures presented here.

In the following sections, we describe the three main steps of the implementation of the protocol schematized in Fig. 1. Steps 1, 2 and 3 are presented in sections 3, 4.2, and 4.3 respectively. In this last section, we also address the question of the linear additivity of the sensitivity experiments. The description and implementation of the passive tracers are addressed in section 5.

## 3 Step 1: reproducing a piControl coupled experiment with an ocean-only configuration

In this section, we describe how to set up the ocean-only CTL experiment (Fig. 1) to reproduce the ocean state of the coupled piControl simulation. The ocean configuration set up for this study is based on the one used in IPSL-CM6A-LR, where we replaced the sea-ice component, as explained above, by ice-ocean fluxes extracted from the coupled model and prescribed to the ocean-only experiment. We also deactivated the biogeochemistry component as we did not intend to look into the biogeochemical impacts. This configuration is forced with outputs from the coupled piControl simulation.

The first stage of the protocol is to select the correct fluxes needed to force the ocean. A synthetic schematic of the interactions between the liquid ocean (under sea-ice) and all the other components of the IPSL-CM6A-LR model is presented in Fig. 2, and the list of needed fluxes is given in Table 1. All these fluxes (heat, freshwater, salt, wind stress), must be outputted from the coupled simulation at high frequency, and used in the forced experiments without time interpolation. We choose to output them at 3-hourly frequency, twice the coupling frequency in IPSL-CM6A-LR, in order to solve the diurnal cycle and remain close to piControl variability. The second stage of the protocol is to verify that the model is conservative, and the heat and freshwater budgets are closed. Moreover, ocean global budgets must be quasi-identical between the coupled piControl and the ocean-only CTL experiments. A slight difference in total incoming heat, freshwater and salt fluxes is however detected (as shown in Table 2) and is solely due to a slightly larger global ocean area in the coupled configuration which includes closed seas.

### 3.1 Freshwater fluxes

The ocean receives and loses water from liquid and solid precipitation, evaporation, sea ice melting and freezing, river runoffs, iceberg melting and iceshelf melting. Under-iceshelf cavities are closed in the configuration of the ocean used here. Instead, in the current version of the model, the mass of water contained in the icesheets is conserved, and all excess precipitation falling on the icesheets eventually returns to the ocean through different melting terms. In the Southern Hemisphere, 50% of the mass goes into iceshelf melting along the coast (based on the Depoorter et al. (2013) climatology), and 50% into iceberg melting along the Merino et al. (2016) climatological map. In the Northern Hemisphere, all the mass goes into a calving term

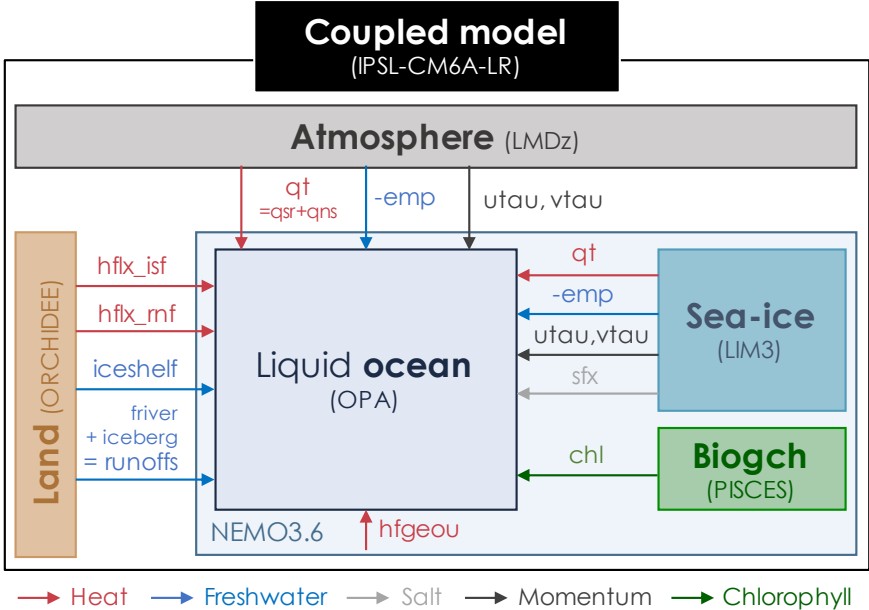

**Figure 2.** Schematic of the exchanges between the ocean and the other components in the IPSL-CM6A-LR coupled model affecting ocean physics. See Table 1 for the signification of these fluxes. LMDz = atmosperic model of the Laboratoire de Météorologie Dynamique, z stands for zoom. ORCHIDEE = ORganizing Carbon and Hydrology in Dynamic EcosystEms. LIM3 = Louvain-la-Neuve Ice Model version 3. OPA = Océan PArallélisé. PISCES is a biogeochemistry model; it is not an acronym. NEMO = Nucleus for European Modelling of the Ocean.

uniformly distributed over the Northern Hemisphere ocean. The freshwater flux from rivers, icebergs and iceshelf melt are vertically distributed over a prescribed depth. Table 1 presents how these terms are grouped and defined in the model.

The net amount of water entering the ocean liquid interface is given by -emp + runoffs + iceshelf (Fig. 2). If the model is conservative, for a given time period $\Delta t$, the global ocean volume change (in m³ s⁻¹) should verify:

$$\frac{\Delta V}{\Delta t} = \frac{1}{\rho_0} \times \overline{\sum_{i,j}((-\text{emp}(i,j,t) + \text{runoffs}(i,j,t) + \text{iceshelf}(i,j,t)) \times \text{areacello}(i,j))}^t \tag{1}$$

Where $\rho_0$=1026 kg m⁻³ is the ocean volumic mass of reference, areacello is the ocean grid cell area, and $i$ and $j$ are the horizontal coordinate indices. All three terms emp, runoffs and iceshelf outputted from the piControl are needed to force the ocean. Averaged over the global ocean (see Figs. 3a and Table 2), the main balance is between emp and friver (1718 mSv and 1636 mSv resp. in CTL, averaged over the entire simulation). The contribution from iceshelf melt and icebergs (the latter being included in runoffs) are very small (41 mSv each; 1 mSv = 10³ m³ s⁻¹). The net incoming freshwater flux for the simulation period is ∼1 mSv, equivalent to a total volume change of ∼9086 km³ of water over the simulation. As compared to the mean volume of ocean water ($1.33 \times 10^{12}$ km³) this change can be considered close to zero.

| Short name | Signification | Unit | CMIP6 equivalent |
|---|---|---|---|
| **emp** | Evaporation-Precipitation, includes sea-ice formation and melt, and calving in the NH | kg m$^{-2}$ s$^{-1}$ | **-(wfo+friver+ficeberg)** |
| **runoffs** | River runoffs + iceberg melting ($>$ 0 into ocean) = friver + iceberg | kg m$^{-2}$ s$^{-1}$ | **friver+ficeberg** |
| **iceshelf** | Iceshelf melting ($>$ 0 into ocean) | kg m$^{-2}$ s$^{-1}$ | **flandice** |
| **qt** | Net downward heat flux = qns + qsr | W m$^{-2}$ | **hfds** |
| **qsr** | Downward shortwave flux | W m$^{-2}$ | **rsntds** |
| qns | Downward non solar heat flux (includes hflx_icb and hflx_cal) | W m$^{-2}$ | nshfls |
| **hflx_rnf** | Sensible heat flux from river and iceberg runoffs (at SST) | W m$^{-2}$ | **hfrunoffds** |
| hflx_icb | SH iceberg latent heat loss (<0), included in qns | W m$^{-2}$ | |
| hflx_cal | NH calving latent heat loss (<0), included in qns | W m$^{-2}$ | |
| hflx_isf | Heat flux from iceshelf melting (sensible+latent) | W m$^{-2}$ | |
| hfgeou | Geothermal heat flux (constant in time) | W m$^{-2}$ | |
| **sfx** | Downward salt flux into sea water | g m$^{-2}$ s$^{-1}$ | **sfdsi** |
| **utau** | Surface downward x stress | N m$^{-2}$ | **tauuo** |
| **vtau** | Surface downward y stress | N m$^{-2}$ | **tauvo** |
| DCHL | Mass concentration of diatoms expressed as chlorophyll | mg m$^{-3}$ | |
| NCHL | Mass concentration of other phytoplankton component expressed as chlorophyll | mg m$^{-3}$ | |
| **CHL** | Mass concentration of all phytoplankton expressed as chlorophyll = DCHL + NCHL | mg m$^{-3}$ | |
| **siconc** | Sea-ice fraction | $\varnothing$ | |
| **sithic** | Sea-ice thickness | m | |

**Table 1.** Exchanges between the ocean and the other components in the IPSL-CM6A-LR coupled model and their signification. The terms in bold are those read from the coupled model outputs to force the stand-alone ocean model in a fixed-flux configuration. The corresponding CMIP6 terms are given in the right column, indicating the terms needed to compute the flux perturbations.

To check the closure of the freshwater budget over the period 1850-2100, the left hand term and right hand term in equation 1 are computed separately (Table 2). $\Delta V$ is computed as the difference in global ocean volume $V$ between the last and first day of the simulation for CTL, and the last and first 3-hours in piControl. The right hand term in equation 1 is computed from annually-averaged freshwater fluxes. The ocean freshwater budget closes almost perfectly in piControl and CTL, with mismatches of only 0.002 mSv and 0.016 mSv respectively, which corresponds partly to precision errors introduced during data analysis. A larger mismatch in CTL is due to the use of daily means versus 3-hourly means in piControl to estimate $\Delta V$. Indeed, we quickly lose precision in estimating the differences in global scalars when using larger time averages. For example, the use of monthly means instead of daily means to estimate $\Delta V$ in CTL leads to an equivalent freshwater flux of 0.990 mSv instead of 1.128 mSv, leading to an error of 0.154 mSv on the closure of the freshwater budget instead of 0.016 mSv. Furthermore, piControl and CTL have quasi-identical net incoming freshwater fluxes (1.148 mSv vs 1.144 mSv respectively),

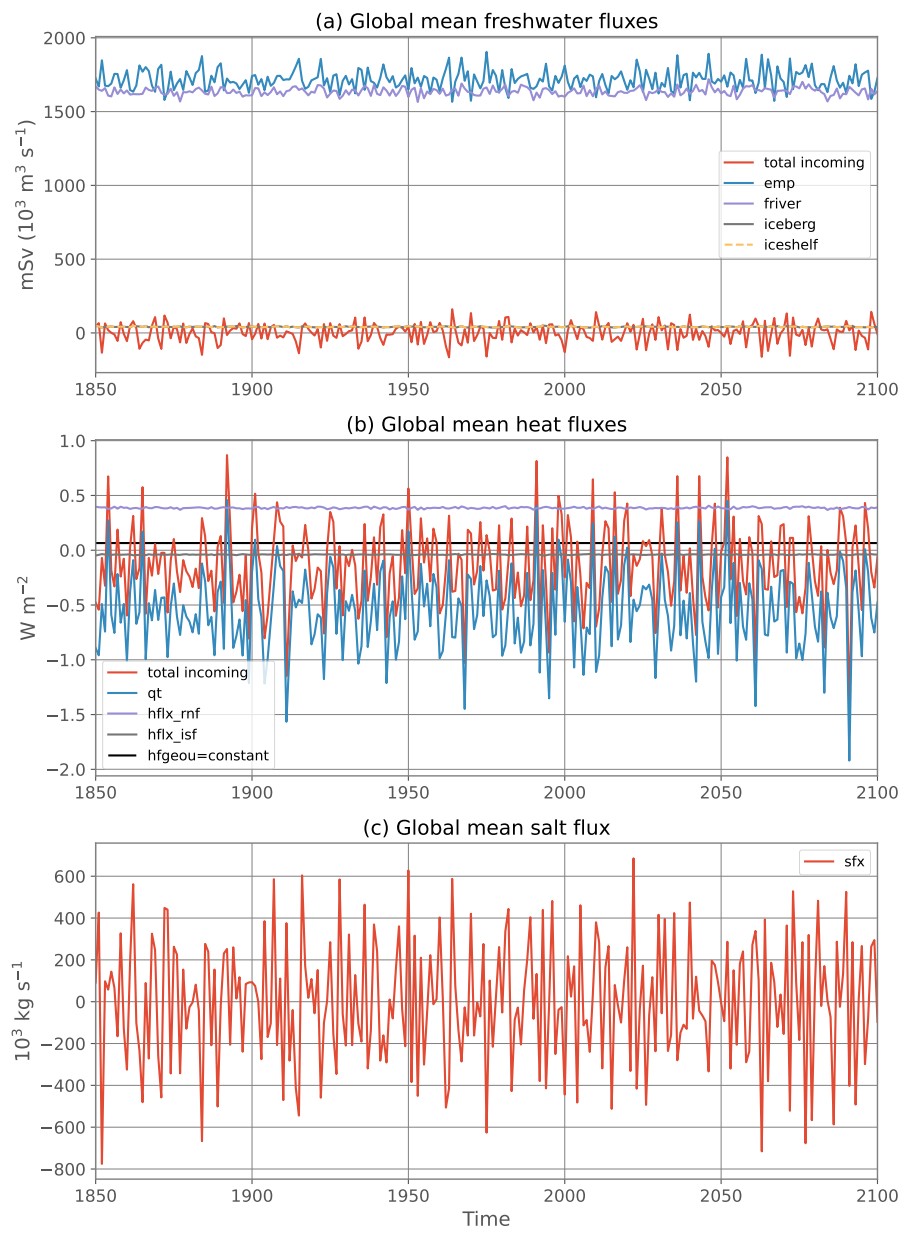

**Figure 3.** Evolution of the globally averaged, annual mean (a) freshwater fluxes, (b) heat fluxes and (c) salt flux in the piControl and CTL experiments. See Table 1 for the signification of each flux component.

the difference being due to the global ocean area which includes closed seas in the coupled configuration while it does not in the stand-alone ocean configuration, as stated before.

|  | piControl | CTL | ALL |
|---|---|---|---|
| emp | 1734.78 mSv | 1717.61 mSv | 1808.35 mSv |
| runoffs | 1694.57 mSv | 1677.41 mSv | 1769.50 mSv |
| iceshelf | 41.34 mSv | 41.35 mSv | 44.27 mSv |
| **Total incoming freshwater flux** | **1.148 mSv** | **1.144 mSv** | **5.426 mSv** |
| **= -emp + runoffs + iceshelf** | | | |
| **vs. $\Delta V$** | **1.146 mSv** | **1.128 mSv** | **5.417 mSv** |
| qt | -0.548 W m$^{-2}$ | -0.546 W m$^{-2}$ | 0.166 W m$^{-2}$ |
| qsr | 172.125 W m$^{-2}$ | 172.114 W m$^{-2}$ | 171.703 W m$^{-2}$ |
| qns | -172.673 W m$^{-2}$ | -172.661 W m$^{-2}$ | -171.537 W m$^{-2}$ |
| hflx_rnf | 0.388 W m$^{-2}$ | 0.386 W m$^{-2}$ | 0.424 W m$^{-2}$ |
| hflx_isf | -0.040 W m$^{-2}$ | -0.040 W m$^{-2}$ | -0.043 W m$^{-2}$ |
| hfgeou | 0.066 W m$^{-2}$ | 0.066 W m$^{-2}$ | 0.066 W m$^{-2}$ |
| **Total incoming heat flux** | **-0.1345 W m$^{-2}$** | **-0.1344 W m$^{-2}$** | **0.6125 W m$^{-2}$** |
| **= qt + hflx_rnf + hflx_isf + hfgeou** | | | |
| **vs. $\Delta OHC$** | **-0.1343 W m$^{-2}$** | **-0.1346 W m$^{-2}$** | **0.6128 W m$^{-2}$** |
| **sfx** | **620 kg s$^{-1}$** | **622 kg s$^{-1}$** | **13 653 kg s$^{-1}$** |
| **vs. $\Delta SC$** | **565 kg s$^{-1}$** | **111 kg s$^{-1}$** | **13 770 kg s$^{-1}$** |

**Table 2.** Freshwater, heat and salt budgets over 1850-2100 in the piControl, CTL and ALL experiments (see Table 1 for the signification of each flux component). All fluxes are globally-averaged. Global ocean changes (volume, heat and salt contents) are converted to flux units for comparison.

## 3.2 Heat fluxes

The ocean exchanges heat at its upper boundary with the atmosphere and sea-ice components, through shortwave radiation (the part not used to melt sea-ice), long-wave radiation and all other non-radiative fluxes (sensible and latent heat from evaporation, precipitation and ice thermodynamics). It also receives sensible heat from river runoffs, sensible+latent heat from iceberg melting in the Southern Hemisphere, latent heat from calving in the Northern Hemisphere and sensible+latent heat from iceshelf melting. As for the freshwater fluxes, the heat fluxes associated with runoffs and iceshelf melting are distributed on the vertical. Finally, the ocean bottom is warmed up by a constant, spatially-varying geothermal heating (Goutorbe et al., 2011). These terms are synthetized in Table 1 together with notations used in the model.

The total heat input into the ocean is qt + hflx_rnf + hflx_isf + hfgeou (Fig. 2). For a given time period $\Delta t$, the global ocean heat content change (in Watts) should verify:

$$\frac{\Delta OHC}{\Delta t} = \overline{\sum_{i,j}((\text{qt}(i,j,t) + \text{hflx\_rnf}(i,j,t) + \text{hflx\_isf}(i,j,t) + \text{hfgeou}(i,j)) \times \text{areacello}(i,j))}^t \quad (2)$$

With $OHC(t) = \rho_0 \times cp \times \sum_V \theta(t)dV$; $cp \approx 3981$ J K$^{-1}$ kg$^{-1}$ the ocean specific heat; $\theta$ the ocean conservative temperature with grid cell volume $dV$. The model needs to read qt, qsr, hflx_rnf and hfgeou to force the ocean. hflx_isf is reconstructed online from the freshwater flux term (iceshelf, see Table 1) and the freezing point temperature. In the NEMO code, the shortwave radiation qsr needs to be specified separately as it penetrates in the top hundred meters of the ocean depending on the chlorophyll concentration field (see subsection 3.5.2 on chlorophyll prescription).

Globally-averaged, the main balance is between solar (qsr=172.11 W m$^{-2}$ in CTL) and non-solar (qns=-172.66 W m$^{-2}$ in CTL) heat fluxes. They almost compensate and yield a net downward heat flux qt=-0.55 W m$^{-2}$ on average (Table 2). The runoffs sensible heat flux (hflx_rnf) and iceshelf heat flux (hflx_isf) partly compensate qt with mean values of 0.39 W m$^{-2}$ and -0.040 W m$^{-2}$ respectively, though their interannual variability is much weaker than qt (see Fig. 3). The net incoming heat flux at the ocean interface sums up to -0.13 W m$^{-2}$ during the 251 years (Table 2), illustrating the disequilibrium (or "drift") found in the piControl of this model, even after 2000 years of simulation (Silvy, 2022). By evaluating the left and right-hand side terms of the heat budget (equation 2) separately, we find a perfect closure of the budget, with an error of 0.002 W m$^{-2}$ both in the piControl and CTL (Table 2).

### 3.3 Salt flux

When the ocean is coupled to a sea-ice model, there is a salt flux exchanged between the two components, as the ice salinity is different from zero: when ice melts, there is a downward flux of both freshwater and salt. This salt flux sfx is thus needed to correctly reproduce the piControl with the ocean-only configuration. For a given time period $\Delta t$, the global ocean salt content change (in g of salt s$^{-1}$) should verify:

$$\frac{\Delta SC}{\Delta t} = \overline{\sum_{i,j}(\text{sfx}(i,j,t) \times \text{areacello}(i,j))}^t \quad (3)$$

With $SC(t) = \rho_0 \times \sum_{i,j,k} S(i,j,k,t)dV$ ; S the ocean absolute salinity (in g kg$^{-1}$) with grid cell volume $dV$.

The globally-averaged salt flux oscillates around zero during the entire simulation (Fig. 3). It has a mean value of $1.7 \times 10^{-9}$ g m$^{-2}$ s$^{-1}$, equivalent to 622 kg s$^{-1}$ integrated over the ocean surface, or to a total of $4.9 \times 10^{12}$ kg of salt exchanged over the simulation period. Evaluating the left-hand term in equation 3 separately (see Table 2), we find an equivalent ocean salt content change rate equal to 565 kg s$^{-1}$ in the piControl and 111 kg s$^{-1}$ in the CTL. These numbers in fact correspond to very small variations compared to the interannual standard deviation of the globally-averaged salt flux of around 300 000 kg s$^{-1}$. Furthermore, the left-hand term is very sensitive to the computation method (e.g. frequency of the global scalar outputs), which can explain the small discrepancy between the left and right-hand terms, but otherwise the closure of the salt budget is respected.

### 3.4 Wind stress

Wind stress is prescribed to the ocean via its zonal (utau) and meridional (vtau) components (Table 1). In ice-covered region, it is the stress below sea-ice that is read (Fig. 2).

### 3.5 Other inputs needed in the ocean-only configuration

#### 3.5.1 Vertical mixing

The parameterization of ocean vertical mixing depends on sea-ice concentration and thickness in the IPSL-CM6A-LR configuration, in particular the mixing length scale. Thus, both the sea-ice fraction *siconc* and thickness *sithic* are extracted from piControl at 3-hourly frequency and used in CTL experiment. Sensitivity tests using sea-ice outputs at lower frequency (typically monthly) led CTL to diverge from piControl in the first months of the simulations (not shown).

#### 3.5.2 Chlorophyll field

The shortwave heat flux penetrates through the surface layers of the ocean. The penetration of that flux in the ocean is modulated by the concentration of total chlorophyll, the only biogeochemical component that has an effect on the ocean physics in NEMO (Fig. 2). When the biogeochemistry model (PISCES, Aumont et al. (2015)) is activated (which is the case in the coupled piControl experiment), the chlorophyll concentration comes from PISCES. When PISCES is deactivated (which is the case for our forced experimental setup), the solar radiation vertical profile in the ocean depends on a prescribed chlorophyll field.

There are 3 options to prescribe a chlorophyll field in the ocean-only configuration of NEMO, plus a fourth one that we implemented specifically for this work:

– option 1: imposing a constant and uniform chlorophyll field at the surface (=0.05 mg.m$^{-3}$),

– option 2: reading a 2D surface file,

– option 3: reading a 2D surface file and reconstruct vertical chlorophyll profiles,

– option 4 (new option implemented): reading a full 3D file which gives chlorophyll concentration everywhere in the water column.

Two options give very satisfying results in terms of SST and vertical temperature profile: reading the surface chlorophyll field from the piControl at monthly frequency with no reconstruction on the vertical (option 2, purple line in Fig. 4), and reading the full 3D chlorophyll field from piControl, also at monthly frequency, (option 4, red line in Fig. 4). The other options make CTL diverge from piControl (options 1 and 3: grey and blue lines respectively in Fig. 4), with some clear vertical redistribution of temperature compared to piControl as shown in panel b. We choose to use option 4 (reading the 3D field) since it is the most accurate way to reproduce the reference piControl.

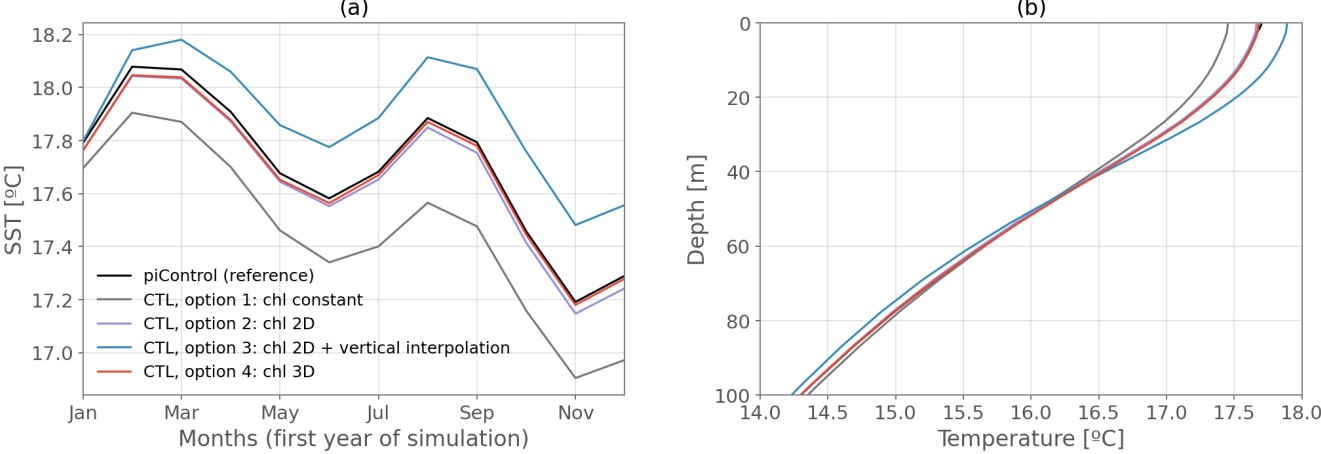

**Figure 4.** (a) Monthly globally-averaged SST the first year of the simulation, and (b) Annual-mean globally-averaged temperature profile in the first 100m, for the: coupled piControl (black), CTL with the 3D chlorophyll read from the piControl (red), CTL with the 2D chlorophyll read from the piControl and interpolated vertically (blue), CTL with the 2D chlorophyll read from the piControl and the same values imposed in the subsurface (purple), and CTL with a constant and uniform chlorophyll value (grey).

We chose to deactivate the ocean biogeochemistry model. However, for other studies it might be interesting to keep it activated to investigate changes in biogeochemical variables in the sensitivity experiments. In this case, the chlorophyll does not need to be prescribed as it is directly read from the biogeochemical model.

### 3.5.3 Temperature below freezing point

As indicated in section 2, the interactive sea-ice model was replaced by prescribed fluxes in the forced configuration. This choice was made to ensure the ocean sees exactly the fluxes exchanged with sea-ice in the coupled model. This is consistent with our main modeling goal which is to reproduce the internal variability of the coupled model in the forced configuration.

However, this setup may lead the temperature to locally fall below freezing point in polar regions, in all the ocean-only simulations including CTL. To deal with this issue, we implemented part of the solution proposed by Todd et al. (2020) in our experiments. Namely the ocean temperature is capped to the freezing point in the equation of state and in the calculation of the Brünt-Vaisala frequency. Hence, heat is conserved in the model while ocean density is capped so as to avoid spurious convection events. In that configuration, temperature can still fall below freezing point. In the ALL simulation for example, it can reach as low as -7ºC in polar areas because of the negative heat flux perturbation especially in the Arctic (see section 4). This represents nevertheless only very small areas of the ocean and does not directly affect the circulation because it does not affect the equation of state, but heat transport can still be affected. We thus tested two alternative methods by constraining the temperature to the freezing point to confirm this choice (see Appendix A). The unconstrained case presented here turned out to better reproduce the coupled model than the alternative methods, both in amplitude and timing of key diagnostics, for the CTL

and ALL simulations, apart from the surface Arctic Ocean. Furthermore, the unconstrained case allows a cleaner comparison with the passive temperature tracer. Consequently, this method is better suited to investigate the timing of physical changes in the ocean interior, their emergence from background climate variability and their attribution to different forcings, as compared to the other methods presented in Appendix A.

### 3.6 Diagnostics and validation

The validation of the CTL experiment is an important step of this work, before going forward with adding the perturbation components (section 4). Since the goal of the ocean-only CTL experiment is to reproduce the ocean state of the piControl, here we compare these two simulations, i.e. their mean state and interannual variability, for a number of different diagnostics.

We keep in mind that CTL is forced at 3-hourly frequency using outputs from piControl. This frequency was chosen for the physics in CTL to remain very close to the piControl. The coupling frequency in the coupled model between the ocean and atmosphere is 1.5 hours (Boucher et al., 2020), so small errors are still introduced in the CTL all along the simulation and can be amplified due to non-linearities of the system. Furthermore, we are also introducing errors by reading the monthly chlorophyll field instead of an interactive chlorophyll and by reading the sea-ice fields at lower frequency than the coupling frequency for the vertical mixing parameterization.

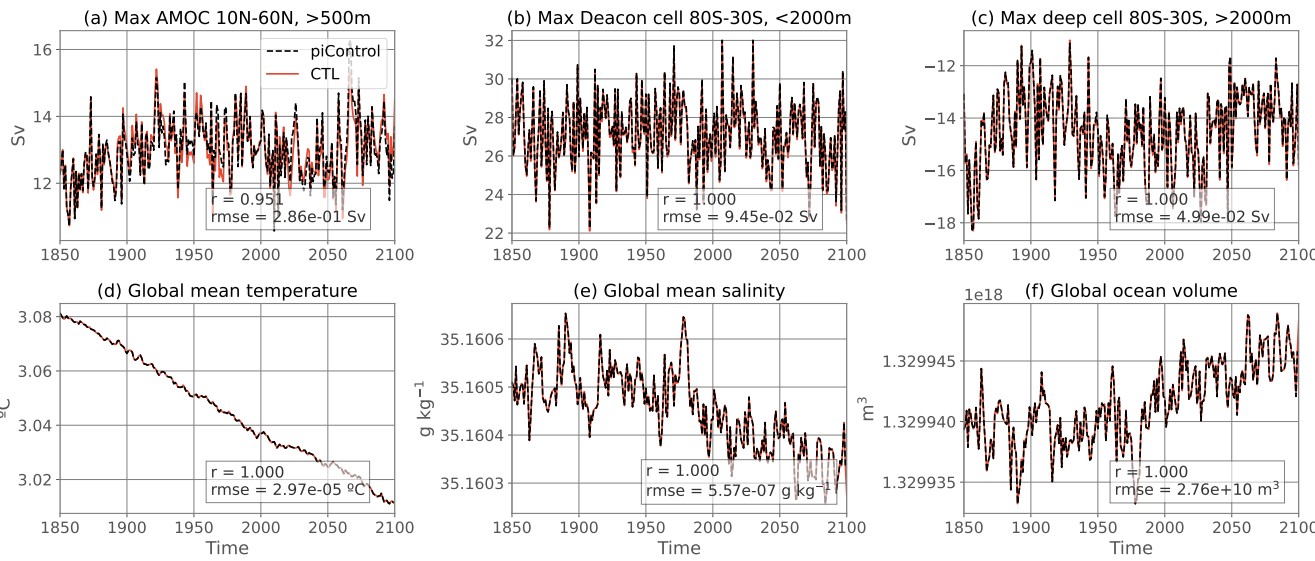

**Figure 5.** Annually-averaged global and overturning diagnostic time series for the coupled piControl (black) and the flux-forced ocean-only CTL (red). The values in the bottom right corner correspond to the Pearson correlation coefficient (r) and to the root mean square error (rmse) between the two time series.

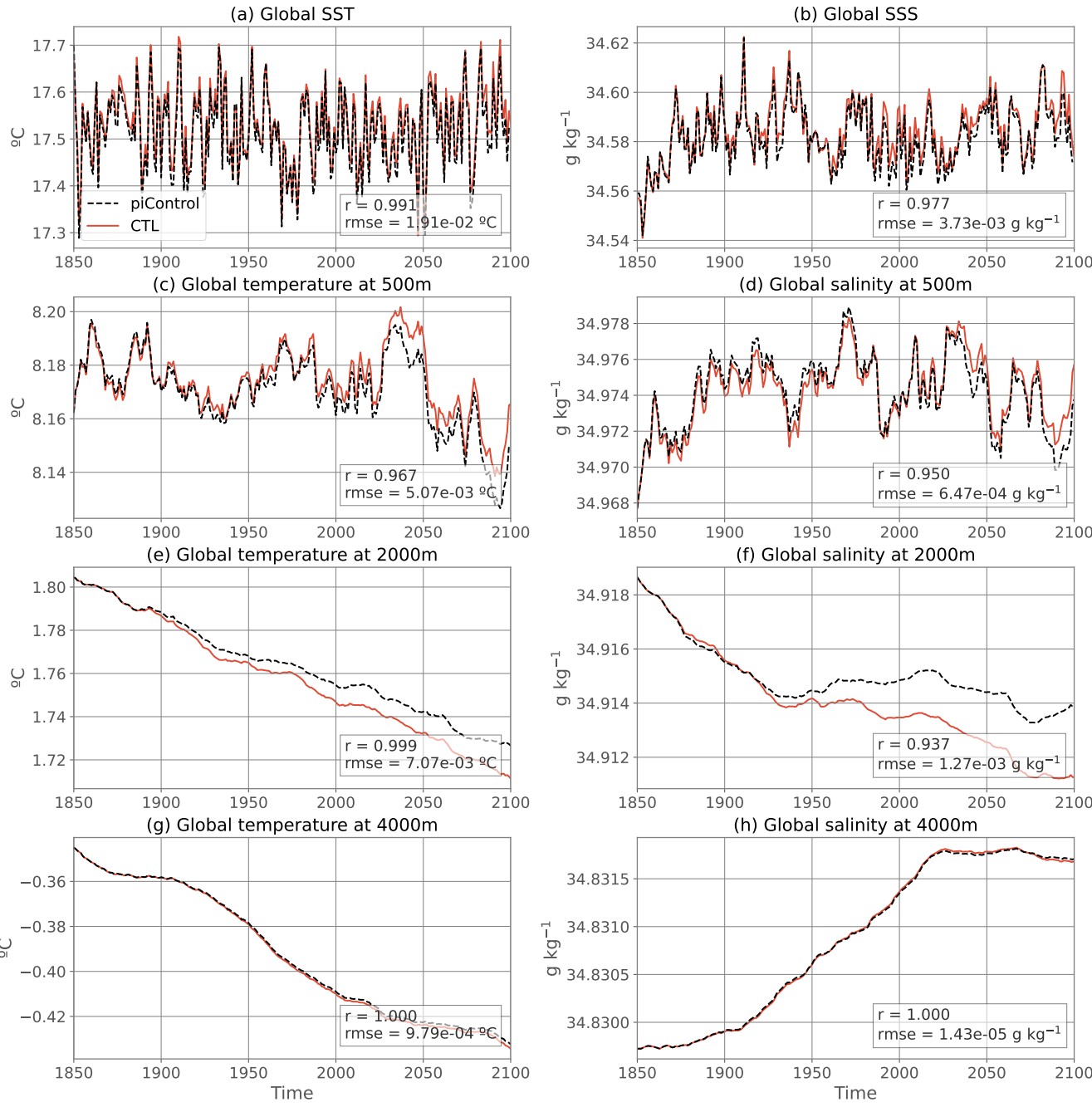

**Figure 6.** As in Fig. 5 but for global mean temperature and salinity at different vertical levels.

Several 1-dimensional diagnostics are presented in Fig. 5 and 6 for piControl (black) and CTL (red) experiments for the
period 1850-2100. In terms of large scale circulation, we show the maximum in Atlantic meridional streamfunction (below

500m, 10°N-60°N, Fig. 5a), as well as the maximum of the global meridional streamfunction for the Deacon cell (above 2000m, 30°S-80°S, Fig. 5b) and deep cell (below 2000m, 30°S-80°S, Fig. 5c) in the Southern Ocean. The similarity between the two experiments is striking during the entire length of the simulations, both in terms of magnitude and of variability, especially for the Southern Ocean cells where the two curves are almost perfectly superimposed (rmse two orders of magnitude weaker than interannual variability). The AMOC time series is also very well reproduced, even though some extremes are not perfectly replicated in terms of amplitude (rmse one order of magnitude smaller than interannual variability).

Global ocean heat, salt and volume are almost perfectly conserved between piControl and CTL (see the budgets in Table. 2) as illustrated by the respective superimposed time series in global mean conservative temperature (Fig. 5d), salinity (Fig. 5e) and total volume (Fig. 5f). In the surface (Fig. 6a,b) and bottom waters (Fig. 6g,h), temperature and salinity variability and mean values in the CTL are the same as the piControl, with small differences in the peak values at the surface. The root mean square error (rmse) is an order of magnitude weaker than the interannual variability. Overall there is no drift away from the piControl even after 250 years of simulation in these variables. However, some small readjustment is slowly appearing at intermediate depths, with warmer and saltier waters at 500m (Fig. 6c,d) and colder and fresher at 2000m (Fig. 6e,f) in CTL than in piControl. The differences are nevertheless very small at the end of the simulation: rmse is about the same order of magnitude as interannual variability. For all the time series presented in Fig. 5 and 6, the Pearson correlation coefficient between CTL and piControl is greater than 0.93, further validating the ocean-only CTL.

We now explore the climatological difference in temperature and salinity at the surface (Fig. 7a,c) and zonally averaged (Fig. 7b,d) between the two simulations. The main differences in SST (Fig. 7a) are localised in the subpolar North Atlantic. This is where slightly below freezing temperatures locally occur in the CTL (not shown) due to the absence of a sea-ice model. The use of a 3-hourly forcing timescale rather than 1.5-hourly (i.e. the coupling frequency in the coupled model) may also induce such differences in this highly sensitive region. The strong SST differences are indeed located in a region of deep convection, where small differences in forcings without any retroaction can cause rapid divergence. These discrepancies however remain mostly limited to the subpolar North Atlantic and they are even smaller by the end of the simulation than at other time periods (not shown). This could indicate strong internal variability governing the differences. In other parts of the globe, we note a warmer Pacific in CTL compared to piControl and cooler Atlantic. Nonetheless, in most parts of the world except for the warmer patch in the subpolar North Atlantic, the differences between the ocean-only CTL and the coupled piControl are smaller than twice the piControl interannual standard deviation and thus consistent with the distribution of interannual variability at the 95% confidence level (represented by the stipples in Fig. 7).

The largest SSS differences (Fig. 7c) are localised in the Arctic with very strong dipoles (>0.5 g kg$^{-1}$ difference) which are however smaller than the piControl interannual variability, are not constant in time (not shown) and seem to stay well within the corresponding ice-covered region of the coupled model. These differences might be due to the sensitivity of surface salinity to the formation and melting of sea-ice and to the fact that the forcing frequency in CTL is not the same as the coupling frequency in piControl, which induces propagating discrepancies. These differences in the Arctic do not impact other areas of the ocean

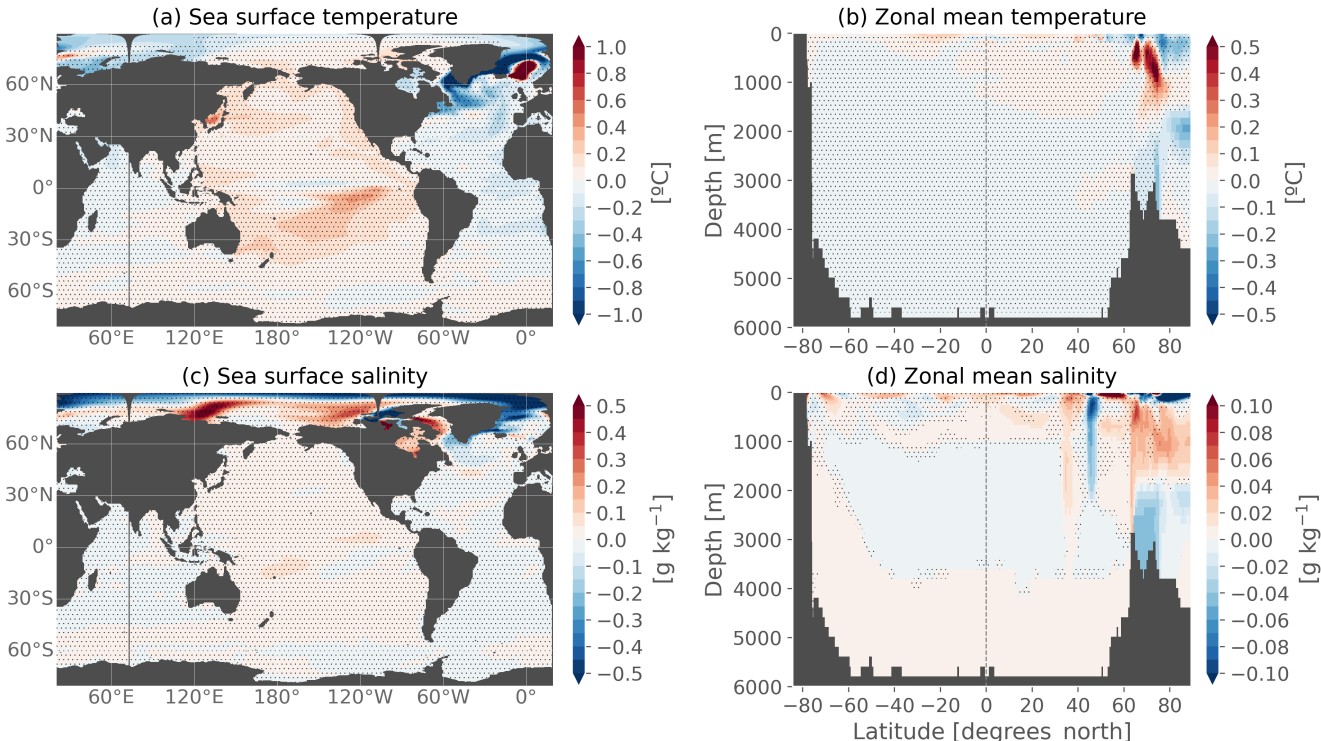

**Figure 7.** Annual-mean climatological differences between the ocean-only CTL and the coupled piControl experiments over the entire simulation 1850-2100 for (a) sea surface temperature, (b) zonal mean temperature, (c) sea surface salinity and (d) zonal mean salinity. Stipples indicate where the difference is lower than twice the interannual standard deviation of the piControl.

where differences remain very small (<0.1 g kg$^{-1}$ and smaller than piControl interannual variability everywhere at the surface) without increasing much in time.

The differences in zonal mean temperature and salinity (Fig. 7b,d) between the CTL and the reference piControl confirm that the largest errors (∼0.2-0.5 ℃ and ∼0.1 g kg$^{-1}$ difference) are located in the Arctic and subpolar regions. The vertical dipolar structures suggest water-mass re-adjustment. There is initially some propagation at depth of these differences especially in the

375 deep convection zone between 60-70 ºN, but the differences don't increase in time after they are installed (not shown). This also confirms that the CTL stays very close to the piControl in all other parts of the ocean, with very small differences between the two experiments (<0.05 ℃ and <0.01 g kg$^{-1}$ outside the surface subtropical gyres). For temperature, these differences are generally smaller than the piControl interannual variability. Localized differences in the Northern high latitudes exceed this threshold because of strongly non-linear dynamics, both for temperature and salinity. For salinity, differences also exceed this

threshold in the global ocean interior, even though the amplitude remains quite small in absolute values (<0.01 g kg$^{-1}$). The reason why temperature is better at reproducing the coupled model in the global interior than salinity remains difficult to be

explained. Most importantly, these differences overall do not expand in time, and the CTL climate is taken as the new reference for the other ocean-only experiments.

## 4 Adding the perturbation components

The CTL experiment gives us the background state and internal variability of the ocean in the IPSL-CM6A-LR coupled model. To reproduce the oceanic response to climate change during 1850-2100 in the coupled model, we now set up the ALL experiment (see Fig. 1) using the same ocean-only configuration as CTL, and add an anomalous component to all surface fluxes forcing the ocean (see Table 1). The anomalous component is constructed as follows:

$$Q'(i,j,t) = Q(i,j,t) - \overline{Q(i,j,t)}^{t=1850-1899}, \tag{4}$$

where the overline denotes a temporal mean, and $Q(i,j,t) = <Q(i,j,t,k)>_{k=1..n}$ is the ensemble mean flux over all available members $k$ of the IPSL-CM6A-LR large ensemble. There are 32 members for all variables over the historical-extended period 1850-2059, and 11 members over 2060-2100, following the ssp245 scenario from 2015; except for iceshelf which was outputted in only 10 members over the full period. Furthermore, the net downward heat flux qt was missing 5 members over the 2015-2029 period. However, that did not affect the consistency of the ensemble means and no discontinuity was found between the
different periods.

Also note that at the time of analysis, there were a few discrepancies in the published outputs of IPSL-CM6A-LR compared to the CMIP6 terminology, namely:

– hfrunoffds (sensible heat flux associated with river and iceberg runoffs) was not included in hfds (hfds=qt), unlike was was suggested in the heat budgets in Griffies et al. (2016).

– wfo (= water flux into sea water) was of opposite sign of what it should have been and was thus positive upward (=E-P-R)

– sfdsi (salt flux) is specified in kg m$^{-2}$ s$^{-1}$ in the CMIP6 requirements, however it was outputted in g m$^{-2}$ s$^{-1}$ in IPSL-CM6A-LR.

Here, we present the orders of magnitude, temporal evolution and spatial patterns of the flux anomalies, before presenting diagnostics to validate the ALL experiment.

### 4.1 Perturbations: budgets and spatial patterns

Table 2 compares the heat, freshwater and salt budgets in the ALL experiment compared to the CTL experiment, which differ only by the addition of the externally-forced perturbations (denoted by a prime sign in the text, e.g. emp').

The evolution of global mean freshwater flux anomalies (Fig. 8a) is a balance between two opposing terms increasing very rapidly at a similar rate (emp' and runoffs'=friver'+iceberg'; both reaching about 300 mSv in 2100), with an additional much
smaller contribution from iceshelf' melting (reaching about 10 mSv in 2100). The latter is however significant in the total

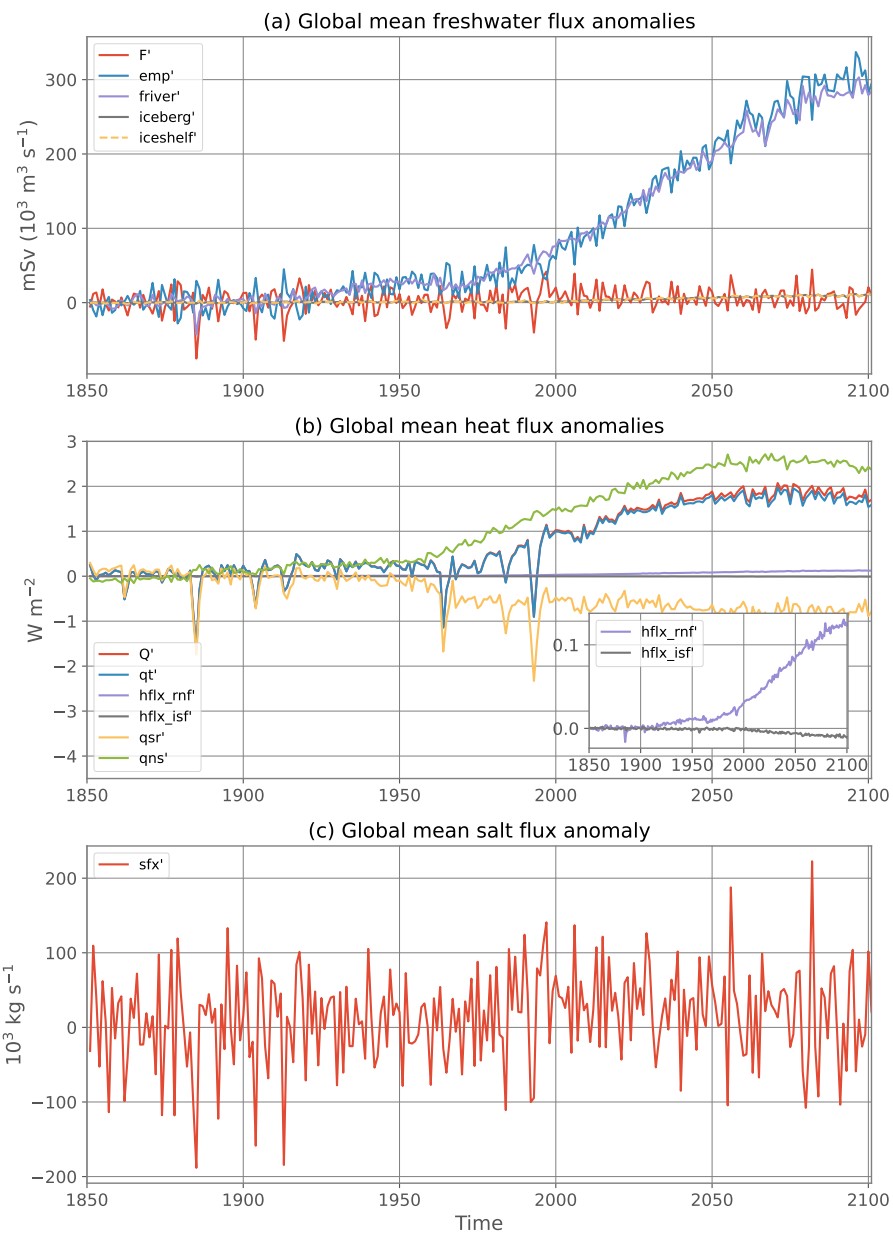

**Figure 8.** Evolution of the globally averaged, annual mean (a) freshwater flux, (b) heat flux and (c) salt flux anomalies as computed from the IPSL-CM6A-LR ensemble mean anomalies relative to 1850-1899. See Table 1 for the signification of each flux component.

balance since the two large terms almost balance each other to yield an order of magnitude similar to the iceshelf' term. The balance between these three terms is similar to the freshwater flux budget in CTL (Table 2). Each individual freshwater flux

anomaly has a lower globally-averaged value than its CTL counterpart. After summation there is a net positive input of water in the ocean by the freshwater flux perturbations.

The increase in global mean surface heat flux anomaly (qt') is dominated by the increase in the non-solar heat flux term (qns') and damped by the decrease in the solar heat flux (qsr'), see Fig. 8b. There is a clear signature of the impact of volcanic eruptions (e.g. 1883, 1963, 1982, 1991) in the qsr' and qt' terms. The anomalous heat flux components from runoffs hflx_rnf' and iceshelves hflx_isf' are more than one and two orders of magnitude smaller than qt', respectively (see inset plot in Fig. 8b). Still, there is a significant increase in hflx_rnf' owing to the large increase in river runoffs (Fig. 8a), but only a small decrease

in hflx_isf'. The IPCC AR6 report assesses a global ocean heat content change of 8.42 [6.08-10.77] ZJ yr$^{-1}$ during the period 1971-2018 (Gulev et al., 2021), equivalent to a 0.74 [0.53-0.94] W m$^{-2}$ heat flux over the ocean area. As a comparison, the total incoming heat flux anomaly in this study during the same period is 0.68 W m$^{-2}$, consistent with the observed assessment.

   Finally, globally integrated, the salt flux anomaly has a small positive trend with a much larger interannual variability (Table 2 and Fig. 8c), hiding marked spatial patterns of opposite signs (not shown).

   The spatial patterns of the anomalies averaged over the 21$^{st}$ century (Fig. 9a,c,e,g) qualitatively largely agree with those of the FAFMIP anomalies (Gregory et al., 2016). We recall that the latter are constructed from a multi-model mean of 1%CO2 idealized experiments, thus we expect differences with the perturbations in the present study due to different external forcings, timescales and models. They are reproduced in Fig. 9b,d,f,h for comparison purposes. This similarity gives confidence in the

response of the IPSL-CM6A-LR model to external forcings relatively to other coupled models in terms of surface fluxes. The most prominent features include large heat uptake over the subpolar North Atlantic and Southern Ocean; enhanced freshwater input over the tropics and high latitudes and weakening over the subtropical gyres, and intensifying and poleward-shifting westerly winds over the Southern Ocean.

## 4.2 Step 2: reproducing the oceanic transient response to climate change

The goal for the ALL experiment is to simulate a climate evolution consistent with the one simulated in the IPSL-CM6A-LR historical+ssp245 ensemble. In other terms, the ALL experiment alone should have a comparable response to any member of this ensemble, i.e. a long-term trend on top of a specific phase of internal variability. It should thus be within or close to the ensemble envelope. Furthermore, the difference between the ALL and the CTL experiments (i.e. the estimate of the forced response in our ocean-only setup) should, in theory, resemble that of the ensemble mean anomaly (i.e. the estimate of the forced

response of the coupled model). Differences between these two estimates of the forced response may however inevitably appear for several reasons. First, we impose the anomalous components at monthly frequencies. Furthermore, iceshelf and to a much lesser extent qt had a few members missing due to problems in the output files on different time periods (see above, section 4). This means that the ocean in the ALL experiment has seen slightly different forcings than those represented by the full ensemble mean oceanic results which we are trying to reproduce. Second, in response to an anomalous surface cooling and in the absence of the sea-ice component and of any retroaction, the ocean cools instead of forming sea-ice, which leads

to below-freezing temperatures in the Arctic (see section 3.5.3 and Appendix A). Third, the absence of sea-ice model also

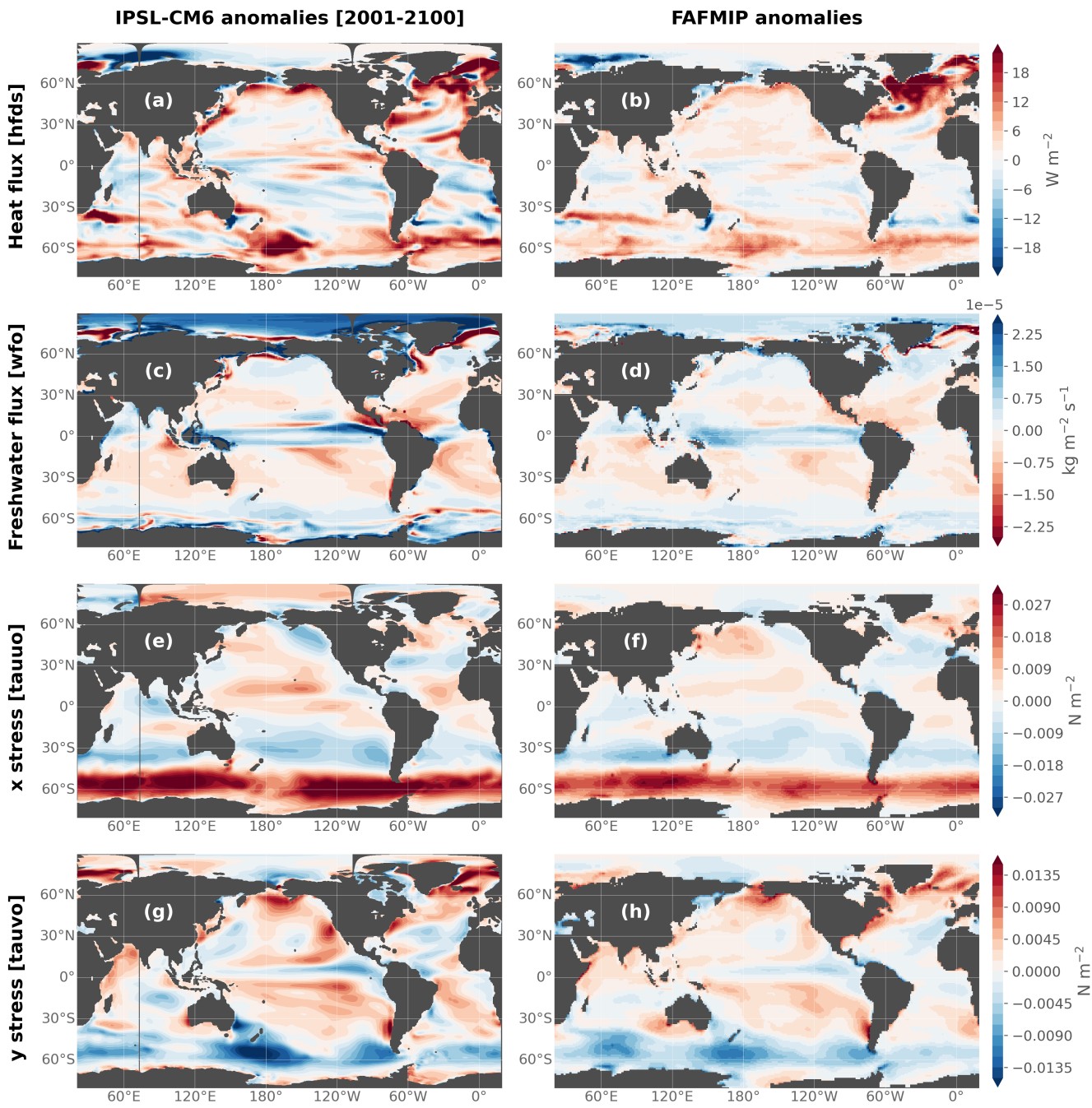

**Figure 9.** Heat flux, freshwater flux and surface downward x and y stress perturbations. (a,c,e,g) IPSL-CM6A-LR ensemble mean anomalies averaged over 2001-2100 relative to 1850-1899 compared to (b,d,f,h) FAFMIP anomalies downloaded from http://www.fafmip.org.

prevents the parameterization of vertical mixing under ice-covered areas to be fully exact. For all the sensitivity experiments, we choose to keep ice fraction and thickness as in the CTL experiment (section 3.5.1). Hence, mixing rates are not dependent on sea-ice melt resulting from climate change (anthropogenic response). This choice is made for consistency reasons since the only external fields that must differ between the sensitivity experiments are the anomalous surface fluxes. Moreover, even though we could have provided the ice information from the IPSL-CM6A-LR historical+ssp245 ensemble for the vertical mixing of the ALL experiment, we cannot provide this information to the HEAT, WATER, STRESS and BUOY experiments since they are idealized cases with no reference for sea ice. The same choice is made for the chlorophyll field, which we keep as in CTL while it certainly changes under anthropogenic forcing.

Nevertheless, even with these limitations, the global response and patterns of change of the ALL experiment very closely resemble that of the IPSL-CM6A-LR historical+ssp245 ensemble. This is illustrated by several global and overturning diagnostics (Fig. 10) showing that the ALL experiment follows the response and stays within or close to the range of the large ensemble during the entire simulation (apart from a few peaks in interannual variability). It has its own initial state and internal variability phased on the ocean-only CTL (and thus on the coupled piControl as well, cf Fig. 5 and 6). Additionally, it contains the response to external forcings inherited from the flux perturbations, marked by long-term increasing anthropogenic emissions. Thus, it acts as an additional individual member: its behaviour compares well to that of individual members of the large ensemble (see e.g. member r1i1p1f1 in Fig. 10, thin orange line). Because of the ocean-only configuration forced with surface fluxes, there is no feedback to the atmosphere when adding a perturbation, thus all our experiments are aligned with the piControl internal variability and can be compared in time (see the coinciding interannual variability between ALL and CTL in Fig. 10). Consequently, the forced response can be diagnosed precisely by taking the difference between the ALL and the CTL experiment. Note that since the internal variability in the ALL experiment is inherited from the piControl simulation, it is not modified by the external forcings, unlike what occurs in simulations of the 21$^{st}$ century in which internal modes of variability can be affected by anthropogenic forcings (Bonnet et al., 2021).

We also compare the forced response in the ALL experiment to the ensemble mean response of the coupled model for zonal mean temperature and salinity anomalies in the mid-21$^{st}$ century (Fig. 11a-f), as well as for vertically-integrated ocean heat and salt content (Fig. 11g-l) and sea surface temperature and salinity (Fig. 12). We show these diagnostics towards the end of the simulation (and at the end of the period over which the large ensemble is available) since it is when the ocean-only framework can most diverge from the coupled model response that we are trying to reproduce, as there are no retroactions in the CTL nor in the ALL experiments, which means differences tend to increase in time.

Before doing that, the data represented in Fig. 11 has to be dedrifted to make correct comparisons. Indeed, there is a cooling drift in the deep ocean in the IPSL-CM6A-LR model (see the heat budget of the piControl in Table 2, consistent with Mignot et al. (2021)). This drift is also visible in Fig. 10a. It leads to 1) different initial states between the historical members due to the fact that they start from different dates of the pre-industrial control simulation and 2) contamination of the externally-forced trends if the data is not corrected for drift. Dedrifting is conducted for both temperature and salinity outputs by fitting a

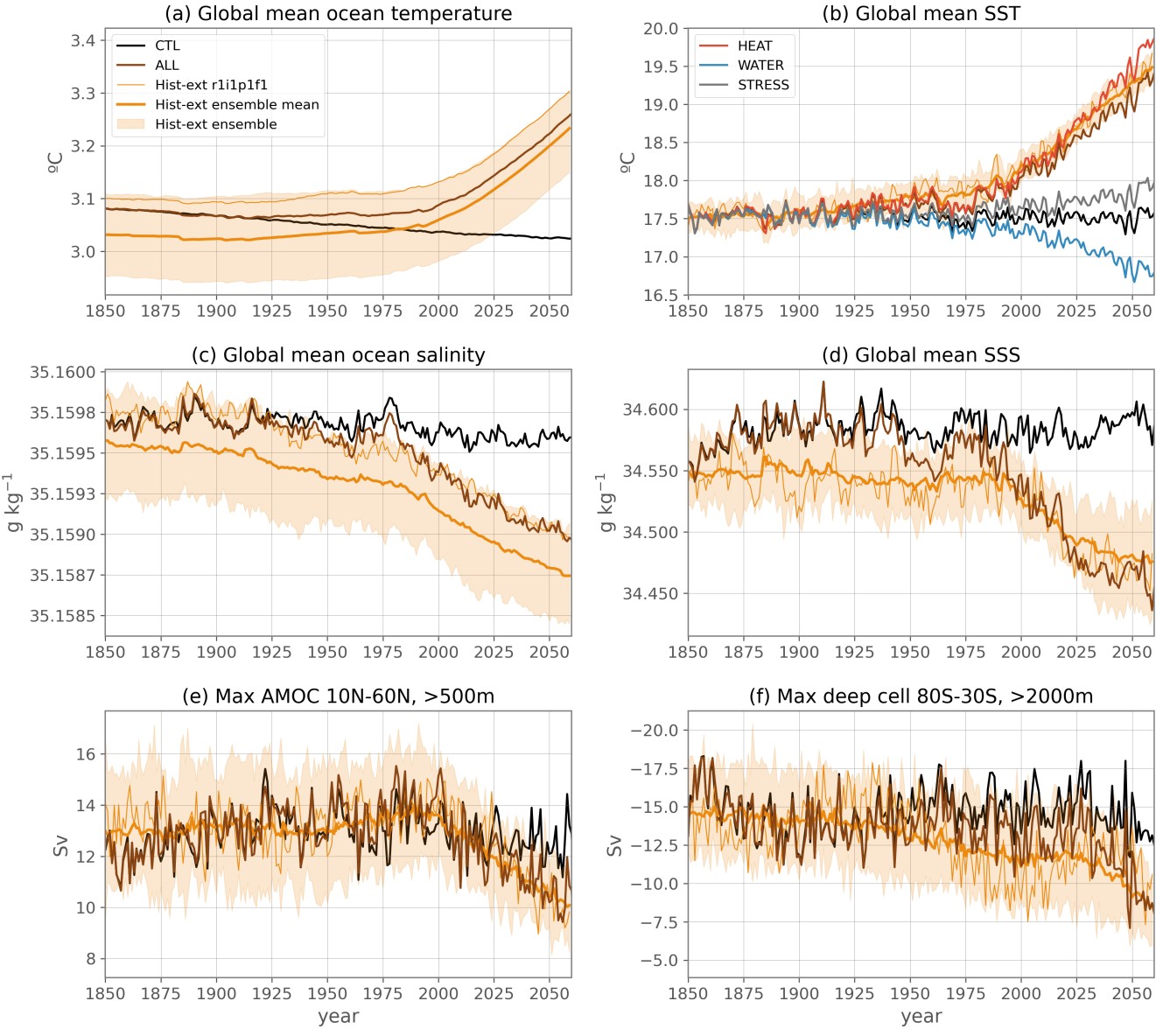

**Figure 10.** Annually-averaged diagnostic time series for the flux-forced ocean-only CTL (black), ALL (brown) and the historical-extended large ensemble (orange shading = intermember spread; bold orange line = ensemble mean; thin orange line = one individual member). In panel b, the SST is also represented for the ocean-only sensitivity experiments HEAT (red), WATER (blue), and STRESS(grey). Panels a,c represent averages over the entire, full-depth, ocean volume.

second-order polynomial to the 2000-year piControl at each gridpoint and removing the corresponding period of this fit from the historical members as well as from the ocean-only simulations which inherit an identical drift. The same mean state is then

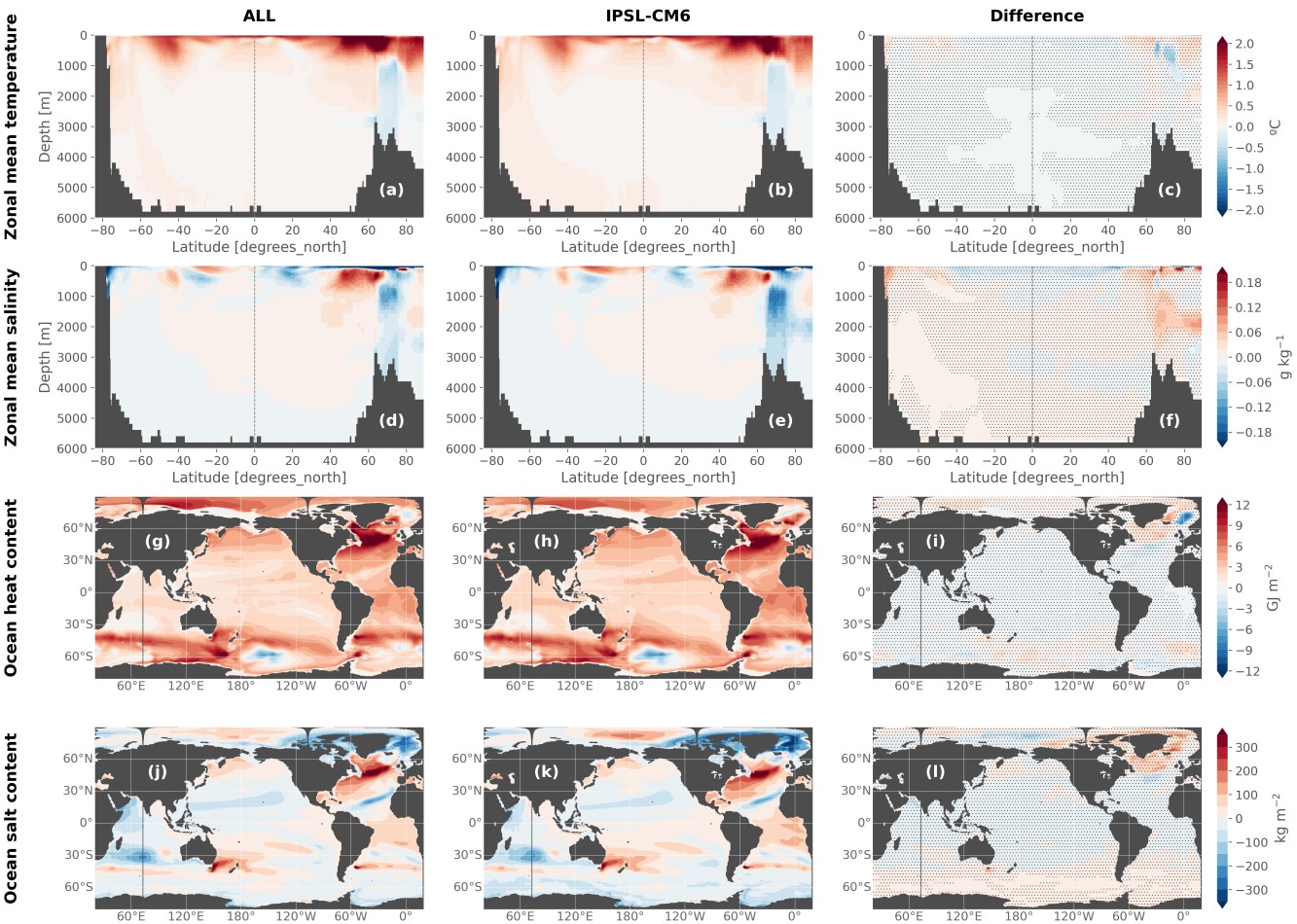

**Figure 11.** Annual-mean anomalies in zonal mean temperature (a-c), zonal mean salinity (d-f), vertically-integrated ocean heat content (g-i) and ocean salt content (j-l), averaged over 2040-2059 relative to 1850-1899. The anomalies are calculated for the forced response in the ALL experiment (i.e. the difference between ALL and CTL; left column) compared to the forced response in the IPSL-CM6A-LR model as calculated from the ensemble mean (middle column). The difference between the ocean-only forced response in ALL and the coupled model ensemble mean are represented in the third column; the stipples indicate where the difference is lower than twice the intermember standard deviation. All the data has been corrected for ocean drift in this figure.

added to the historical members so that their initial conditions only differ because of internal modes of variability. Note that
omitting this dedrifting step leads to artificially very large intermember spread due to different initial states.

The response of the ALL experiment for zonal mean temperature, salinity, heat and salt contents is strikingly similar to the response of the IPSL-CM6A-LR ensemble mean, as displayed in Fig. 11. The difference between the two is smaller than the ensemble spread as measured by twice the intermember standard deviation almost everywhere, and as represented by the

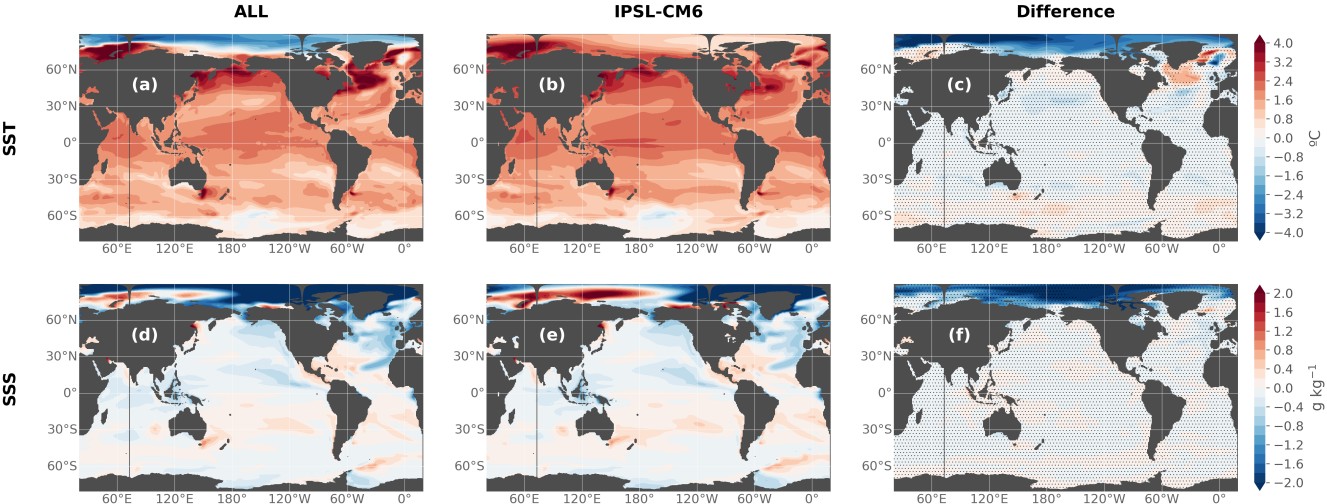

**Figure 12.** Same as Fig. 11 but for sea surface temperature (a-c) and salinity (d-f).

stippled areas in the right column of Fig. 11. This close reconstruction of the forced response validates the coherence of this
ocean-only configuration.

The only notable discrepancy seen for the SST anomaly is in the Arctic (Fig. 12) where surface temperatures fall below the
freezing point in the ALL experiment (due to the absence of the sea-ice component), creating negative anomalies (Fig. 12a)
unlike in the IPSL-CM6A-LR historical+ssp245 response where there is a small warming (Fig. 12b). This problem is addressed
in Appendix A. We show that by constraining the temperature to not fall too much below the freezing point, we can solve these
negative anomalies and have better SST anomalies in the Arctic, but at the cost of degrading the temperature anomalies in other
parts of the ocean.

The SSS patterns in the Arctic are also affected (Fig. 12c-f), with a larger freshening in the ALL experiment compared to
the coupled model, particularly North of the Bering Strait. Nonetheless, these strong discrepancies in surface temperature and
salinity are only located in the Arctic surface layers and don't seem to impact the rest of the ocean surface, nor the ocean
interior as illustrated by the previous diagnostics (Fig. 11).

When averaging SST and SSS everywhere outside of the Arctic (Fig. 13), we can see that the forced response of the ALL
experiment (brown line) reproduces well the forced response of the coupled model (orange line) at every time step. A small
difference is seen in the 21st century for SST but does not grow much in time, and stays smaller than the intermember range.
The general behaviour and time evolution of the coupled model is thus well reproduced by the ocean-only framework for SST
outside the Arctic. For SSS, this difference is larger than SST compared to the total change and this remains to be investigated
(see also section 3.6). These discrepancies might be very region-dependent depending on dynamical regimes more or less
affected by the absence of ocean-atmosphere or ocean-ice feedbacks (e.g. the evolution in time of the coupled model forced

temperature response is better reproduced in upper ocean mode waters than in deep and abyssal water-masses, (Silvy et al., 2022)).

More generally, we note the Arctic is the region of the world where we have the most difficulties reproducing the response of the coupled model, both in the CTL (Fig. 7) and the ALL (Fig. 12) experiments, due to the absence of an interactive sea-ice model in our simulations. Interestingly, the absence of an interactive sea-ice model is found to be much less problematic in the Southern Ocean (although we do note some significant differences in salinity in the deeper parts, Fig. 11f,l). This work is thus not designed to study the mechanisms at play in the Arctic Ocean in the IPSL-CM6A-LR model, and any result in this region
should be interpreted carefully.

In all other regions, we have managed to reproduce satisfactorily the IPSL-CM6A-LR large ensemble response with an ocean-only model and can coherently decompose the individual flux anomalies to investigate the different physical mechanisms within this framework.

### 4.3 Step 3: decomposing the oceanic transient response with sensitivity experiments

The technical validation of the sensitivity experiments consisted in verifying whether the heat, freshwater and salt budgets were consistent with the CTL or the ALL experiment. In other words, in the simulations that have the heat flux anomalies activated, the heat budget should be identical as in the ALL experiment, while simulations that do not have the heat flux anomalies should have the same heat budget as CTL. These budgets were indeed verified and are rigorously equal to those shown in Table 2. Furthermore, the heat, freshwater and salt budgets are also equal to those in ALL/CTL (depending on the experiments) at each
time step, i.e. global mean temperature, salinity and volume are overlaid during the entire simulation (not shown).

In terms of scientific validation, we verified whether:

1. the sensitivity experiments were phased in terms of internal variability, as explained in the goals of the study in section 2, so as to extract the estimate of the externally-forced component by taking the difference from CTL;

2. the linear-additivity hypothesis was correct, i.e. is the response to all the forcings (ALL) equal to the sum of the responses to individual forcings (HEAT+WATER+STRESS or BUOY+STRESS);

3. our results were comparable to other similar studies in the scientific literature in terms of patterns of long-term change.

To address point 1, an example is shown in Fig. 10b for SST, in which we can see that ALL, HEAT, WATER and STRESS are well phased with CTL in terms of interannual variability when ignoring their respective long-term trends caused by the
climate change perturbations. This is also the case for the other variables, which we do not show here for readability.

Point 2 is partly addressed in Fig. A1 of Silvy et al. (2022), showing that for temperature, differences between ALL and HEAT+WATER+STRESS are mostly significant towards the end of the simulations in polar regions, where strongly non-linear interactions take place. By comparing [HEAT+WATER+STRESS - ALL], [BUOY+STRESS - ALL] and [HEAT+WATER - BUOY], we can trace which flux interactions cause these non-linearities. For example, we show that non-linearities in the

subpolar North Atlantic are created by the interaction of heat and freshwater flux perturbations, while wind stress perturba-
       tion plays a minor role; on the contrary, non-linearities in the subpolar Southern Ocean arise from the interaction of all three
       surface flux perturbations. We add time series of SST and SSS anomaly (Fig. 13), comparing ALL (brown) with the sum
       HEAT+WATER+STRESS (dotted black). These two curves remain close together during the entire period, and differences,
       suggesting non-linearities, are much smaller than the intermember spread (orange shading). This demonstrates that the decom-
position of processes in ALL using these sensitivity simulations can be made confidently. As shown in Silvy et al. (2022),
       regions of deep convection are the least additive. This indicates that the attribution of ocean changes to individual surface per-
       turbations cannot be interpreted quantitatively in these regions as accurately as in other regions. Nonetheless, this mechanistic
       decomposition is still a useful tool to understand the physical causes of ocean changes.

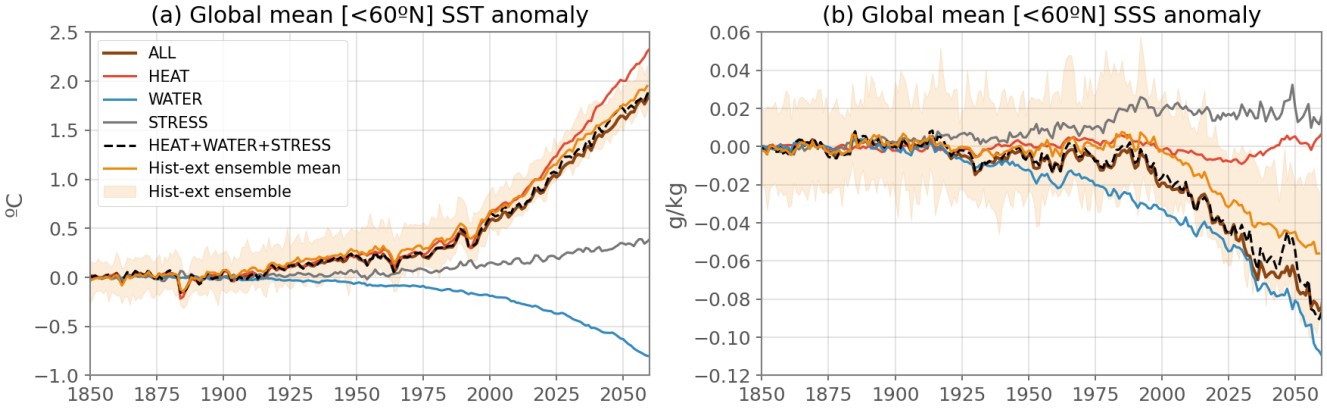

**Figure 13.** Evolution of global mean sea surface temperature (a) and salinity (b) anomalies, excluding the Arctic, for the ocean-only ALL
simulation (brown), HEAT (red), WATER (blue), STRESS (grey), the sum of HEAT, WATER and STRESS (dotted black), and the coupled
model historical ensemble mean (orange). The ensemble mean anomaly is taken as the difference at each time step from the 1850-1899
average. The anomalies in the ocean-only experiments are computed at each time step as the difference between each experiment and CTL,
before removing the 1850-1899 average, to be consistent with the ensemble mean anomaly.

       Finally, for point 3 we compared the long-term response of these sensitivity experiments to what was done previously in
the scientific literature in similar numerical designs (e.g. Fyfe et al. (2007); Armour et al. (2016); Gregory et al. (2016); Liu
       et al. (2018); Shi et al. (2020); Todd et al. (2020)). Note however that most studies focused on heat storage and not on salinity.
       Here, we show the long-term change in ocean heat content in HEAT, WATER and STRESS in Fig. 14a,c,e (also see Silvy et al.
       (2022) Fig. 4 for zonal mean temperature). Although the amplitude, timescale and implementation of the forcings are very
       different, these results are very coherent qualitatively to previous studies cited above in terms of spatial patterns. The common
patterns are described in more details in Silvy et al. (2022). We show furthermore the similarity of our results to the FAFMIP
       multi-model mean of Couldrey et al. (2021) in Fig. 14. While keeping in mind that the amplitudes of ocean heat content
       change cannot be directly comparable between the two studies (different forcings, timescales and configurations), and that
       we are looking at single model versus multi-model mean results, the patterns of change in HEAT, WATER and STRESS very

closely match those of the FAFMIP results. Differences are expected from varying details in the protocols, but the similarity

with FAFMIP and with other studies at the first order scientifically validates the long-term response of these simulations. This enables us to investigate the mechanistic decomposition of the historical+scenario response to anthropogenic climate change and its time evolution which were left unexplored in past modeling designs.

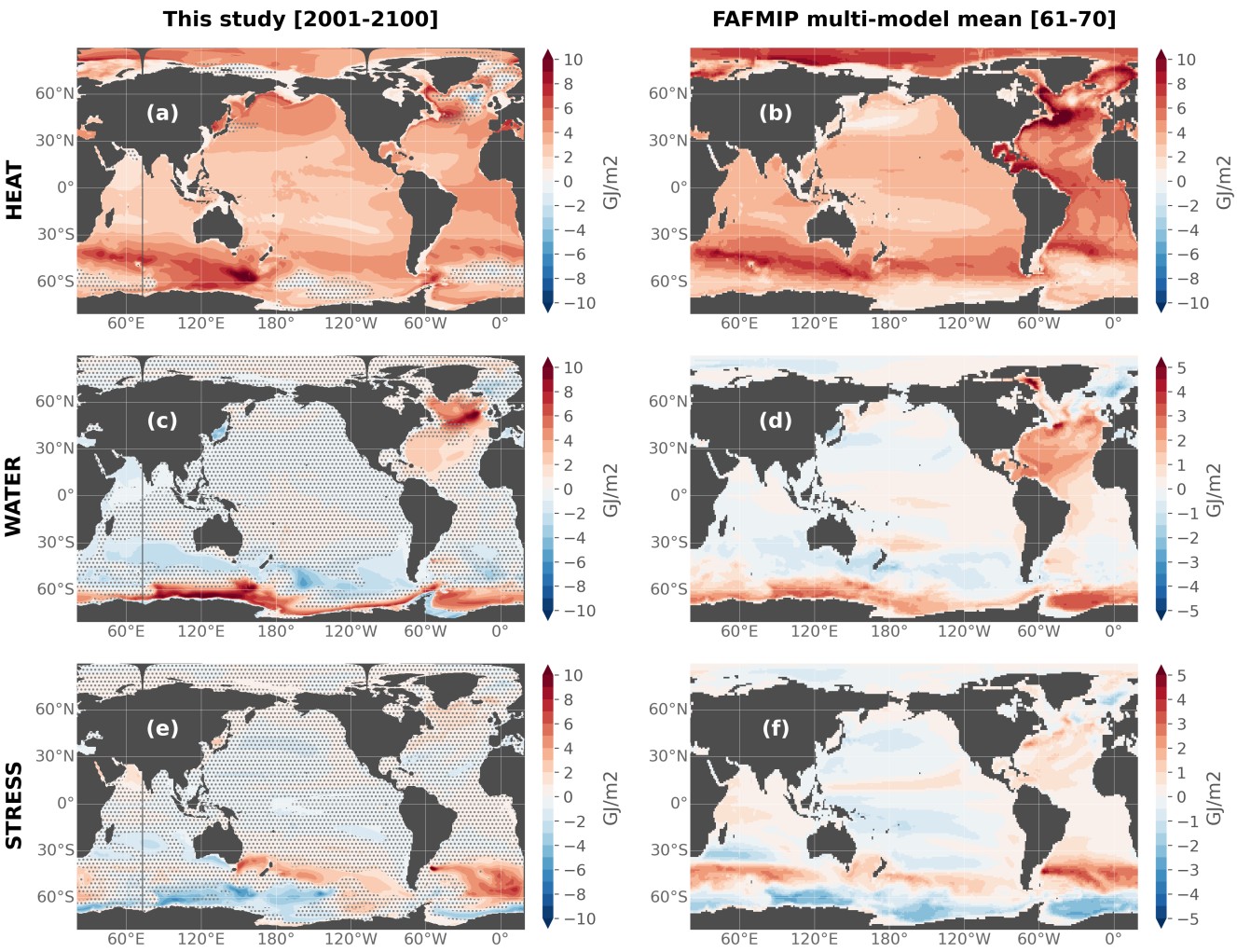

**Figure 14.** Vertically-integrated ocean heat content anomalies in the HEAT, WATER and STRESS experiments in this study (a,c,e) and in the FAFMIP multi-model mean of Couldrey et al. (2021) (b,d,f, which correspond to the data in their Fig. 6a-c). The mean is performed over 10-12 global climate models (see details in Couldrey et al. (2021)). Anomalies are averaged over the period 2001-2100 for this study, and years 61-70 for the FAFMIP simulations. In a,c,e, stipples indicate where the anomaly is lower than twice the interannual standard deviation of the CTL experiment.

## 5 Implementing the passive tracers

In order to perform in-depth analysis of temperature and salinity evolution in response to climate change, two passive tracers are implemented using the TOP component in NEMO3.6 (Tracers in the Ocean Paradigm). These tracers will allow to diagnose the temperature and salinity anomalies in the perturbed experiments due solely to the addition of heat or freshwater in the ocean, without the contribution from the redistribution of pre-existing heat and salt in response to circulation changes. In the following, we present their design based on Banks et al. (2002) and Banks and Gregory (2006), and their implementation in the code.

### 5.1 Passive Anomaly Temperature

First, we write a simplified equation of the evolution of the prognostic temperature in the model by using a similar terminology as Gregory et al. (2016) and Couldrey et al. (2021):

$$\frac{\partial T}{\partial t} = Q + \Phi(T). \tag{5}$$

Q is the net heat flux at the ocean boundaries (=qns+qsr+hflx_rnf+hflx_isf+hfgeou, we omit the constants $\rho_0$, $cp$ and the grid cell thickness by which Q needs to be divided for homogeneity), and the operator $\Phi$ represents advection and parameterizations of sub-grid scale processes for temperature. For simplification, $\Phi$ is called transport in the following.

Second, we split T, Q and $\Phi$ into a CTL (unperturbed), and anomalous (perturbed) component, so that $T = T_{CTL} + T'$, $\Phi = \Phi_{CTL} + \Phi'$, and $Q = Q_{CTL} + Q'$. The evolution of the temperature anomaly T' then writes as:

$$\frac{\partial T'}{\partial t} = Q' + \Phi_{CTL}(T') + \Phi'(T') + \Phi'(T_{CTL}). \tag{6}$$

The evolution of T' thus depends on the anomalous surface heat flux Q' (=qns'+qsr'+hflx_rnf'+hflx_isf'), also called "added heat", the anomalous transport of CTL temperature ($\Phi'(T_{CTL})$), also called "redistributed heat", and the transport of T'.

Third, we further decompose T' into a passive uptake of added heat and a redistribution of pre-existing heat, so that $T' = T'_a + T'_r$. $T'_a$ (for added heat), also called PAT (for Passive Anomaly Temperature), is implemented as a passive temperature tracer representing the transport of added heat in the ocean without affecting the dynamics of the ocean. It is expressed in ºC, initialized to 0, forced by the anomalous heat flux Q' (similarly as T') and transported in the ocean by the full circulation $\Phi = \Phi_{CTL} + \Phi'$ as follows:

$$\frac{\partial T'_a}{\partial t} = Q' + \Phi_{CTL}(T'_a) + \Phi'(T'_a) \tag{7}$$

Since there is no feedback on the surface fluxes in our protocol, all the excess heat Q' that enters the ocean acts entirely to change the global ocean heat content, that is, over a period of time $\Delta t$,

$$\iint \overline{Q'}^{\Delta t} dA = \frac{\rho_0 cp}{\Delta t} \iiint T' dV = \frac{\rho_0 cp}{\Delta t} \iiint T'_a dV = \frac{\Delta OHC}{\Delta t}, \tag{8}$$

with dA a surface grid cell area and dV a grid cell volume. The redistributed temperature, diagnosed from $T_r' = T' - T_a'$, has no effect on the global ocean heat content and only changes temperature locally:

$$\frac{\rho_0 cp}{\Delta t} \iiint T_r' dV = 0, \tag{9}$$

All the forcing terms are applied to the passive tracer trend in the same way as for temperature: qns' acts on the top ocean level; qsr' penetrates into the ocean with the same absorption coefficient as for qsr; hflx_rnf' and hflx_isf' are distributed vertically.

Since $T_a'$ does not affect the dynamics of the ocean, we can implement it in all the ocean-only simulations using the same forcing term Q', irrespective of whether the perturbations are applied on the prognostic temperature (ALL, HEAT, BUOY) or not (CTL, WATER, STRESS). The only difference in the evolution of $T_a'$ between the ocean-only simulations is its transport by the circulation which is specific to each experiment. In CTL, $\Phi' = 0$ by definition, so $T_a'$ becomes the heat added from externally-forced perturbations but passively transported by the CTL circulation, which corresponds to the passive tracer of *faf-passiveheat* from the FAFMIP protocol (Gregory et al., 2016). It is also comparable to the temperature change in climate change simulations for which the circulation is constrained to its climatological state (Winton et al., 2013; Bronselaer and Zanna, 2020). Consequently, in CTL we have $\frac{\partial T_a'}{\partial t} = Q' + \Phi_{CTL}(T_a')$ and we can decompose the evolution of the added heat $T_a'$ in the ALL experiment into:

$$\left. \frac{\partial T_a'}{\partial t} \right|_{ALL} \approx \left. \frac{\partial T_a'}{\partial t} \right|_{CTL} + \Phi'|_{ALL}(T_a'|_{ALL}). \tag{10}$$

The difference among the $T_a'$ tracers in-between experiments allows to diagnose the effect of the perturbed circulation on the added heat, that is approximately the 2$^{nd}$-order term $\Phi'(T_a')$.

Thus we can decompose the total temperature change T' in ALL, HEAT or BUOY as $T' = T_a' + T_r'$ (Total = Added + Redistributed). We can further decompose the added component $T_a'$ into an added heat component in the absence of circulation changes ($T_a'|_{CTL}$ or "passive heat" to be coherent with the FAFMIP terminology), and the remaining added heat due to the reorganization of added heat in response to circulation changes ($T_a' - T_a'|_{CTL}$, referred to as "non-linear added heat" since it is a 2$^{nd}$-order term). We can then write $T' = (T_a'|_{CTL}) + (T_a' - T_a'|_{CTL}) + T_r'$, or "Total = Passive + Non-linear added + Redistributed".

Technical validation of PAT is not straightforward since there is no reference to compare it quantitatively in the model. One obvious check is to see whether the integral of PAT is equal to that of T' (Eq. 8), which is verified (not shown). We scientifically validate our PAT tracer by comparing its long-term spatial distribution with previous studies, even though they mostly focused on the response to idealized forcings (e.g. Banks and Gregory (2006); Winton et al. (2013); Marshall et al. (2015); Garuba and Klinger (2016); Gregory et al. (2016); Dias et al. (2020); Todd et al. (2020). We find very similar patterns in most regions (see Silvy et al. (2022) for a more complete description of the common patterns). For instance, the spatial patterns of the different contributions of ocean heat content change in the HEAT experiment (Total, Passive, Non-linear added and Redistributed) match

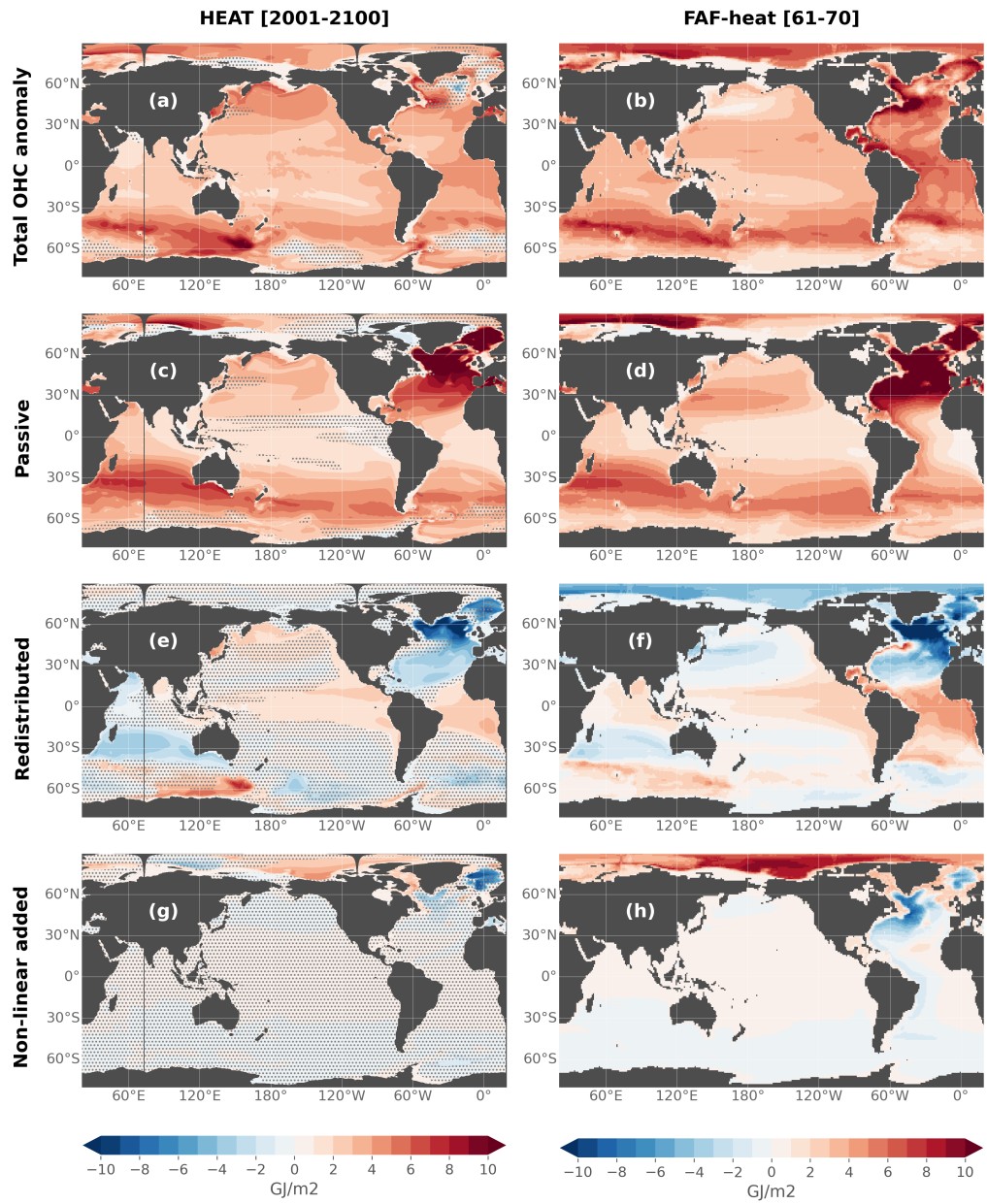

**Figure 15.** Decomposition of total, passive, redistributed and non-linear added heat components (see text for definitions). Shown are the components for vertically-integrated ocean heat content anomalies in the HEAT experiment in this study (a,c,e) and in the FAFMIP multi-model mean of Couldrey et al. (2021) (b,d,f, which correspond to the data in their Fig. 12a,c,e,g). Anomalies are averaged over the period 2001-2100 for this study, and years 61-70 for the FAFMIP simulations. In a,c,e, stipples indicate where the anomaly is lower than twice the interannual standard deviation of the CTL experiment.

fairly well with the FAFMIP multi-model mean of Couldrey et al. (2021) (Fig. 15) although one could expect large differences due to the different forcings and models used in the two studies. This illustrates that the passive temperature tracer behaves as it should in this framework.

## 5.2 Passive Anomaly Salinity

Similarly to temperature, the Passive Anomaly Salinity (PAS or $S'_a$) is forced by the freshwater and salt flux anomalies (the same as the prognostic salinity sees in the perturbed experiments), and can be implemented in all simulations (even CTL) as PAS does not affect the dynamics.

However, the implementation of PAS is more complex than PAT, since the sources and sinks of PAS can be separated unlike for PAT. In NEMO, in particular, when the local ocean volume freely evolves in time (variable volume formulation), the surface boundary condition for ocean salinity is the salt flux (sfx) only. However, ocean salinity is also locally influenced by freshwater fluxes (emp, runoff, iceshelf) through a concentration/dilution effect. Furthermore, for every simulation, PAS must only be affected by the externally-induced anomalies (sfx', emp', runoffs', iceshelf') and not by the background fluxes from the piControl. In particular, we do not want PAS to be impacted by the background freshwater fluxes, present in the concentration/dilution effect. Moreover, the concentration/dilution effect differs between experiments since the freshwater flux perturbations are used to force only some of the experiments (ALL, WATER, BUOY), and not the others (CTL, HEAT, STRESS). Hence:

1. PAS is forced by sfx' in its trend;

2. to compensate the background concentration/dilution effect on PAS, we remove the effect of the piControl fluxes (emp, runoffs, iceshelf, without the anomalies) from the PAS trend;

3. to obtain the same effects of freshwater flux perturbations on PAS between all experiments, we add freshwater flux anomalies in the PAS trend for CTL, HEAT and STRESS.

PAS is initialized to the approximate ocean global mean salinity (34.7 g kg$^{-1}$), and not 0, for the formulation of the freshwater flux anomaly to be correct. This mean value is removed in all post-processing analyses to obtain an anomaly. All the forcing terms are applied to the passive tracer trend in the same way as for salinity: sfx' and emp' act on the top ocean level while runoff' and iceshelf' are distributed vertically.

There is no scientific literature to compare our passive salinity tracer to. In Fig. 16 we show its patterns in CTL (e.g. "Passive" salinity change $S'_a|_{CTL}$), which clearly illustrate in the upper ocean the fingerprint of water-cycle amplification in response to climate change with dry/salty regions getting saltier (subtropics) and wet/fresh regions getting fresher (equatorial and polar regions), as described for example in Eyring et al. (2021).

For both PAT and PAS, we had to locally, and at each time step, set them to 0 and 34.7 respectively (i.e. no variation) in a small and shallow area of the Arctic north of Canada which, unconstrained, was the source of unphysical values rapidly

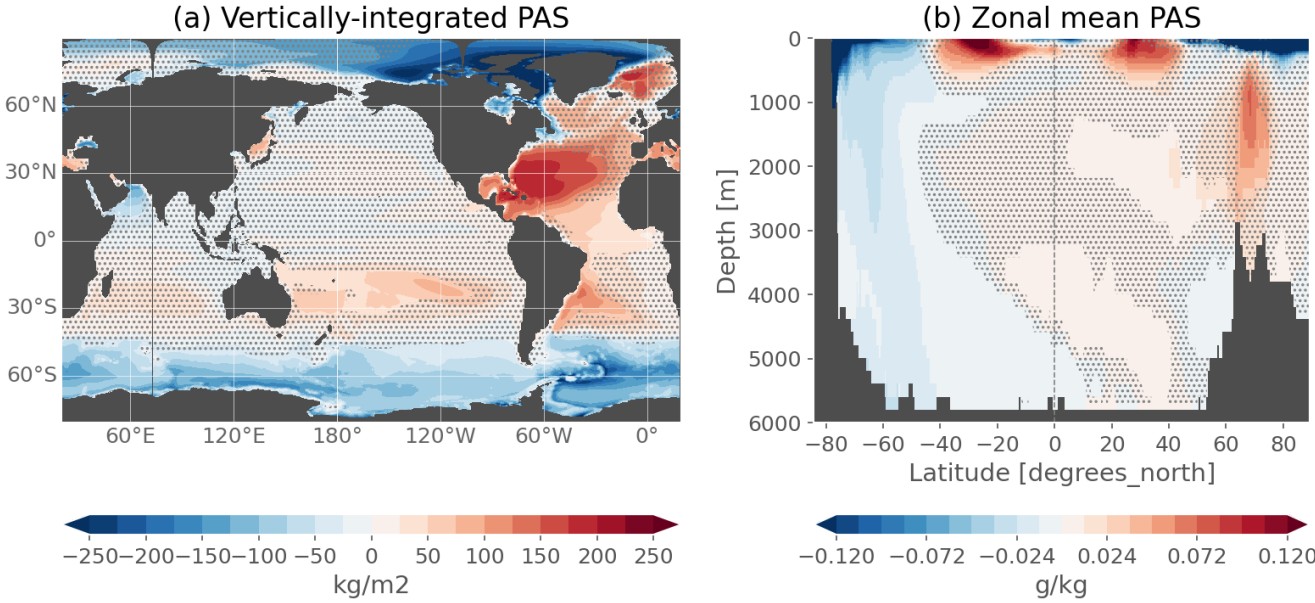

**Figure 16.** Vertically-integrated (a) and zonal mean (b) passive anomaly salinity in years 2040-2059 of the CTL simulations. Stipples indicate where the value is lower than twice the interannual standard deviation of the salinity in the CTL experiment.

propagating out of the area into the Arctic and North Atlantic basins. The reason for this error is yet unknown, although we suspect the shallowness of the Canadian Archipelago could be the source of these large anomalies in tracer values.

## 6    Discussion and conclusions

In this study, we propose a novel modeling framework that aims at untangling the physical mechanisms driving temperature and salinity changes in the ocean interior in response to climate change. We propose a protocol that allows to isolate the
ocean response to the surface flux anomalies in heat, freshwater and winds seen by the ocean during the historical+scenario simulations of a specific climate model. The protocol is however in principle applicable to any climate model and we present its implementation in the context of the IPSL-CM6A-LR simulations of historical and projected human-induced global warming. First, we reproduce a pre-industrial control experiment of the coupled climate model, using an ocean-only configuration of the ocean component of the coupled model. For this, the ocean model is forced with prescribed freshwater, salt, heat and
wind stress fluxes at its boundaries. Second, in a sister simulation, we replicate the ocean's response of the coupled model historical+ssp245 ensemble by adding external forcing perturbations to these fluxes. Third, the role of each of the forcing terms is taken apart by performing sensitivity experiments. Fourth, two passive tracers are implemented to separate the effect of added heat and freshwater from the redistribution of pre-existing heat and salt content by the anomalous circulation.

This paper presents the general approach, describes the implementation of the framework specifically for NEMO3.6, which is the ocean model configuration of IPSL-CM6A-LR, it discusses the choices that are made and validates the simulations. A companion paper presents scientific results on the historical + projected ocean response to anthropogenic climate change based on these simulations (Silvy et al., 2022).

Overall, this framework provides a new way to separate internal variability from the externally-forced signal in the ocean. This is made possible, on the one hand, by a 3-hourly (i.e. high frequency) forcing at the ocean interface with background internally-driven prescribed fluxes, and on the other hand by the extraction of the externally-forced signal from a coupled model large ensemble, which is added to the background fluxes. To ensure that the externally-forced response is extracted correctly, the historical members need to sufficiently sample the various phases of internal variability, and thus a sufficient number of ocean states. Our framework is thus in principle applicable to other large ensembles and their ocean model component, as long as the large ensembles are macro-initialized, i.e. historical members are branched from different times of a long pre-industrial control simulation. Indeed, the historical members are not necessarily sampled similarly in a micro-initialized large ensemble, i.e. where small numerical perturbations are applied in the atmospheric state (Hawkins et al., 2016). The CMIP6 archive could be particularly well adapted since its protocol ensures that the ensembles of historical simulations are macro-initialized, and several modeling centers have performed at least 30 historical members.

Our protocol is inspired by the ocean-only Flux-Anomaly-Forced Model Intercomparison Project (FAFMIP, Gregory et al. (2016); Todd et al. (2020), www.fafmip.org) as well as other similar modeling experiments looking at the role of individual surface flux perturbations (e.g. Mikolajewicz and Voss (2000); Fyfe et al. (2007); Garuba and Klinger (2018); Liu et al. (2018); Shi et al. (2020)). Indeed, the shared goal is to understand the ocean response to the different forcings in response to climate change. However, as explained in the introduction, FAFMIP is by nature a multi-model study and aims at exploring the spread of ocean model responses to the same set of surface flux perturbations. The perturbation fields are constructed from the multi-model mean of idealized simulations in which $CO_2$ concentration increases at a rate of 1% per year, and taken at a time it has doubled in the atmosphere (wrt piControl). These perturbations are constant in time (step-like anomalies with a seasonal cycle but no interannual variations), allowing for long-term responses to be explored at a limited cpu cost (each simulation is run for 70 years). However the transient response to historical climate change cannot be assessed with the FAFMIP protocol. Investigating the evolution in time of the balance of mechanisms causing ocean changes is precisely the novel aspect we have tackled with the present study. Additionally, we propose a framework to explore the mechanistic attribution of ocean changes as effectively seen in individual coupled models during their simulations of the historical period and future projections. This entails implementing exactly the perturbed fluxes as they are seen in the coupled model, as opposed to the generic FAFMIP perturbations. The results obtained from our experiments can be broadly compared with those of FAFMIP simulations in the long-term response, and contain additional information on the transient response during the 1850-2100 period.

As shown by some of the FAFMIP multi-model studies (Gregory et al., 2016; Todd et al., 2020; Couldrey et al., 2021), the spatial patterns of the ocean response to anthropogenic forcings can be model-dependent. Furthermore, the timescale of emergence from internal variability also depends on the model (Silvy et al., 2020). Consequently, a multi-model study following

the protocol presented here would help to explore the uncertainties related to model responses. If other modeling centers were to gain interest in this numerical design and the questions it aims to answer, this would be an interesting inter-comparison exercise, complementing the FAFMIP protocol.

One limitation of the modeling framework presented here concerns the difficulties for the forced configuration to reproduce the coupled model's response to climate change at the surface of the Arctic Ocean. This is probably due to the absence of interactive sea-ice model, although sea-ice dynamics and thermodynamics are taken into account through prescribed ice-ocean fluxes. This choice to not include a sea-ice model is a compromise that allows the best reproduction of the coupled model's internal variability. The protocol could allow additional sensitivity experiments to analyze the effects of sea-ice anomalous fluxes only (not studied here). Another limitation includes non-linear interactions in the system that lead to a non-negligible difference in some regions between the all-forcing simulation (ALL) and the sum of individual-forcing simulations (e.g. HEAT+WATER+STRESS). However this difference is small in most of the ocean.

All the simulations were done on the Jean-Zay supercomputer held at IDRIS center in France (http://www.idris.fr/eng/jean-zay/jean-zay-presentation-eng.html). The main cpu consumption of this protocol is the coupled piControl experiment (430 000 cpu hours), which is needed to output the surface fluxes at high-enough frequency. Running the ocean-only simulations are much cheaper (220 000 cpu hours for the 6 ocean-only experiments). Because this was a novel framework and we had to run many sensitivity tests to the different parameters necessary to force the ocean model, the total cpu time consumed for this work is evaluated at 1 million cpu hours, accounting for 3-4 $tCO_2e$.

*Code and data availability.* The IPSL-CM6A-LR coupled model is presented in Boucher et al. (2020). All the outputs of the CMIP6 experiments (notably piControl, historical and ssp245 used in this study) are publicly available on the Earth System Grid Federation website (https://esgf-node.ipsl.upmc.fr/projects/cmip6-ipsl/). The model code and forcing files are available as follows. LMDZ, XIOS, NEMO, and ORCHIDEE are released under the terms of the CeCILL license. OASIS-MCT is released under the terms of the Lesser GNU General Public License (LGPL). The IPSL-CM6A-LR code is publicly available through svn, with the following command lines: svn co https://forge.ipsl.jussieu.fr/igcmg/browser/modipsl/branches/publications/IPSLCM6.1.11-LR_05012021 modipsl cd modipsl/util; ./model IPSLCM6.1.11-LR. The mod.def file provides information regarding the different revisions used, namely: (1) NEMOGCM branch nemov36STABLE revision 9455; (2) XIOS2 branchs/xios-2.5 revision 1873; (3) IOIPSL/src svn tags/v224; (4) LMDZ6 branches/IPSLCM6.0.15 rev 3643; (5) tags/ORCHIDEE20/ORCHIDEE revision 6592; (6) OASIS3-MCT 2.0branch (rev 4775 IPSL server). The login/password combination requested at first use to download the ORCHIDEE component is anonymous/anonymous. We recommend to refer to the project website: http://forge.ipsl.jussieu.fr/igcmg_doc/wiki/Doc/Config/IPSLCM6 for a proper installation and compilation of the environment. The code modified for the development of the ocean-only numerical experiments with NEMO3.6 is available under: https://zenodo.org/record/6855913. The Python code written to plot the figures of the paper is available under: https://zenodo.org/record/7057865.

## Appendix A: Treatment of the temperature below freezing point: sensitivity tests

As mentioned in section 3.5.3, in both the unperturbed (CTL) and perturbed ocean-only experiments, the prognostic temperature in the ocean model can locally fall below the freezing point since there is no sea-ice component to physically constrain it. Here, we discuss two alternatives to this freely-evolving case by applying different treatments to the temperature when it falls below the freezing point. They are tested on the entire 251 years for both CTL and ALL:

Option 1. At each time step, we constrain the temperature to the freezing point temperature in the mixed layer when it falls below freezing, and we remove the equivalent heat flux in the non-solar heat flux qns distributed over the entire ocean surface, i.e. at each time step and every grid point: $qns(i,j,t) = qns(i,j,t) - qfrz(t)/S$ with qfrz the equivalent heat flux added by blocking the temperature to the freezing point over the mixed layer, integrated vertically and horizontally, and $S$ the ocean area. This redistribution of the added heat flux allows for the global conservation of heat.

Option 2. At each time step, the temperature is relaxed to the freezing point temperature in the mixed layer when it falls below freezing, with a 30-day relaxation period. This is equivalent to adding a positive heat flux in the ocean locally, thus not conserving heat in the ocean globally, nor relative to the coupled model simulations. The question is to know whether this heat flux becomes large in time and if it has an impact on the rest of the ocean.

In the first alternative, the globally-averaged heat flux added and redistributed in qns due to temperatures falling below the freezing point is qfrz=0.12 W m$^{-2}$ in CTL compared to the total incoming heat flux -0.13 W.m$^{-2}$, over the entire simulation (251 years). In ALL, qfrz=0.16 W m$^{-2}$ compared to the total incoming heat flux 0.61 W m$^{-2}$. This additional heat flux is thus globally non-negligible in both simulations and rises over time (not shown). This alternative conserves heat, as shown by the superposition of the red and blue lines in Fig. A1e). However there is a spurious cooling of the surface ocean layers (Fig. A1a) because of the removal and redistribution of qfrz in qns, that is balanced by a warming at depth (Fig. A1c). Salinity also progressively deviates from the piControl at different depths (Fig. A1b and d). The AMOC and Southern Ocean deep cell suffer from the same problem (Fig. A1g and h). In terms of spatial patterns, this alternative prevents the SST from deviating too much from the IPSL-CM6A-LR ensemble mean in the Arctic as expected (compare the freely-evolving simulation Fig. A2a with the alternative simulation Fig. A2d). But it clearly deteriorates the simulation at low and mid-latitudes (compare Fig. A2c and f). Overall this alternative is much less satisfactory than the freely-evolving simulations.

In the second alternative, the additional heat flux from relaxation amounts to 0.074 W m$^{-2}$ in CTL and 0.11 W m$^{-2}$ in ALL. It is less than the redistributed heat in the first alternative but still non-negligible considering over the ALL experiment, qfrz adds an additional 0.11 W m$^{-2}$ to the ocean, on top of the total incoming heat flux of 0.61 W m$^{-2}$. There is a clear spurious warming of the ocean at every depth (Fig. A2a,c,e for CTL; Fig. A2g-i for ALL). Salinity deviates less from the piControl than the first alternative but is still worse than in the freely-evolving simulation (Fig. A1b and d). The AMOC and Southern Ocean deep cell deviations are as strong as in the first alternative (Fig. A1g and h). In terms of spatial patterns, this alternative induces a better SST anomaly compared with the large ensemble in the Arctic as expected (compare Fig. A2a and g). But it clearly

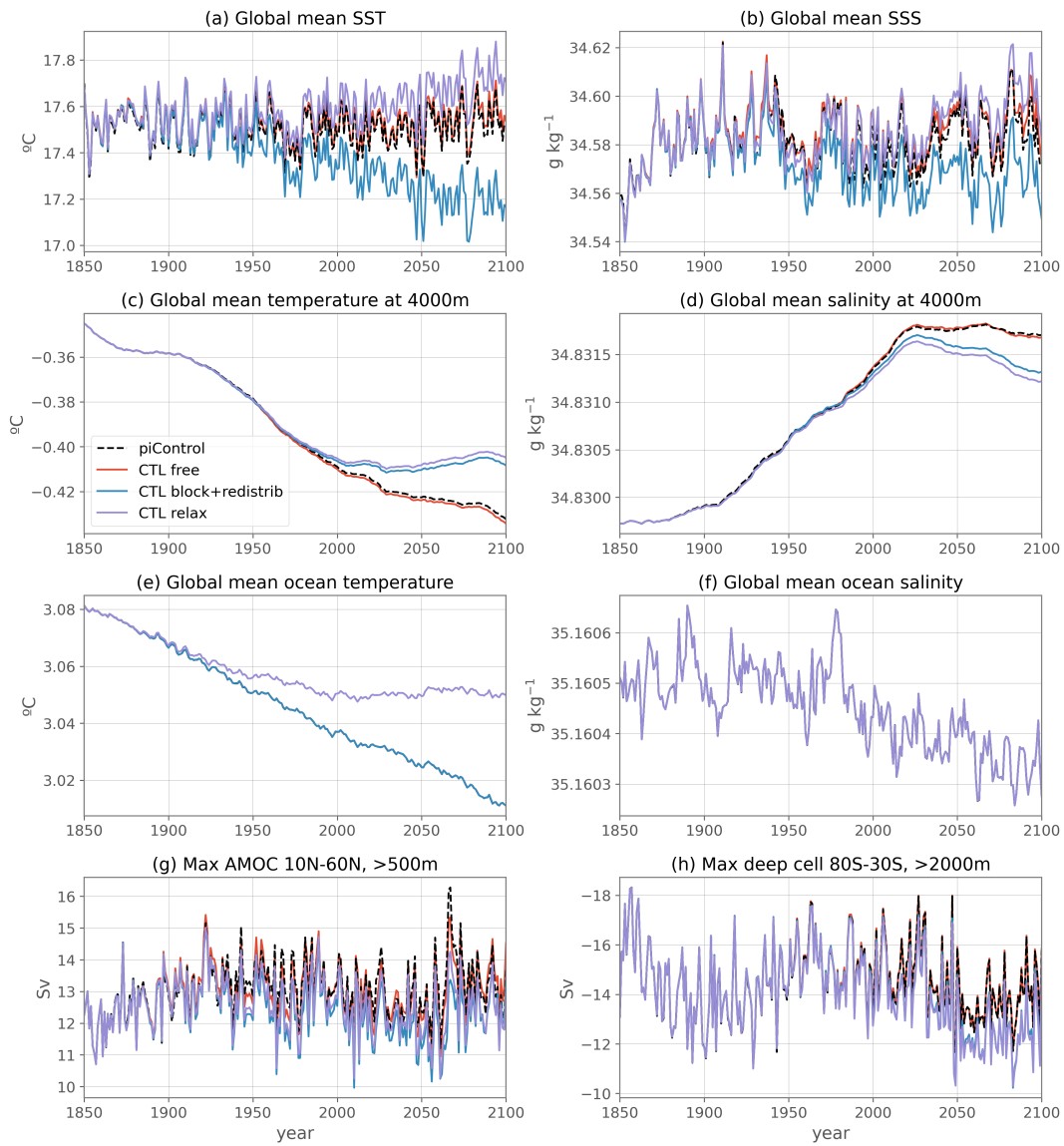

**Figure A1.** Annually-averaged diagnostic time series for the coupled piControl (black); the flux-forced ocean-only CTL as presented in the main text, i.e. in which the temperature evolves freely (red); the CTL in which the temperature is blocked to the freezing point if it falls below it, and the equivalent heat flux that has been added to block the temperature is redistributed over the globe (i.e. removed) in the non solar heat flux (blue, option 1 in the text); and the CTL in which the temperature is relaxed to the freezing point if it falls below it (purple, option 2 in the text). In all three CTL cases, in the equation of state and computation of the Brünt-Vaisala frequency, the temperature is maintained to the freezing point if it falls below it (see section 3.5.3)

.

deteriorates the simulation below the surface layers in the polar regions (compare Fig. A2b and h, and Fig. A2c and i). Finally, the additional heat flux coming from relaxation makes the ocean warm faster and earlier than in the IPSL-CM6A-LR ensemble mean (not shown). This is a big issue when investigating the timing of departure of a warming signal from background climate variability, which is what these simulations are designed for.

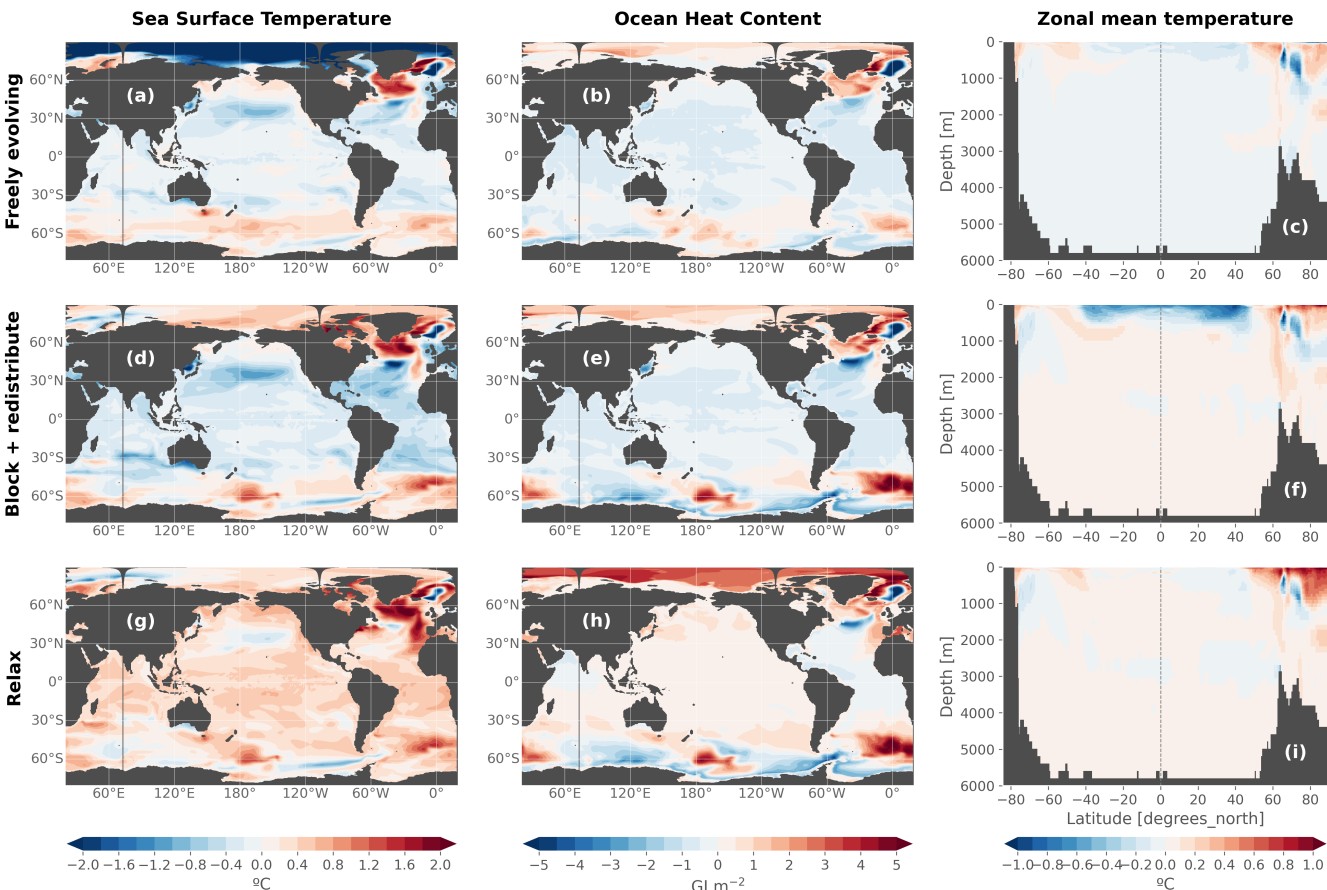

**Figure A2.** Difference between the forced response in 2040-2059 relative to 1850-1899, in the ALL experiment compared to the IPSL-CM6A-LR ensemble mean. The forced response in the different ALL experiments are calculated by subtracting the corresponding CTL experiment, which ensures deep ocean drift is removed. In the IPSL-CM6 simulations, drift was removed prior to computations. These anomalies are calculated for sea surface temperature (left column), vertically-integrated ocean heat content (middle) and zonal mean temperature (right). The three rows compare the ocean-only simulations in which the temperature is allowed to evolve freely (top), in which it is blocked to the freezing point when it falls below it and the associated heat flux is redistributed over the ocean surface (middle, option 2 in the text), and in which it is relaxed to the freezing point when it falls below it (bottom, option 1 in the text).

In summary, apart from better reproducing the SST pattern in the Arctic, and preventing the temperature from falling below the freezing point, these two alternatives do not better reproduce the response of the IPSL-CM6A-LR ensemble mean. Letting the temperature evolve freely is found overall to not impact the response to anthropogenic climate change in other regions than the Arctic and it is without a doubt better at reproducing the temporal evolution of the piControl, which ensures minimal drift from our reference simulation. Furthermore, one of the goal of these experiments is to investigate the evolution of the added and redistributed heat components by implementing a passive anomaly tracer forced with identical surface flux perturbations as the prognostic temperature (see section 5). Modifying the prognostic temperature internally means creating discrepancies in the relationship between temperature change and the passive tracer. We thus chose to run all the sensitivity experiments without any treatment of the temperature below freezing other than in the equation of state and Brunt-Vaisala frequency. The surface Arctic ocean is thus not a region of choice to analyse these experiments and should be considered with care. These technical choices are made in response to our scientific constraints. However, other scientific interests might have led to different choices.

*Author contributions.* Y.S and C.R developed the ocean-only configuration with the help of C.E and G.M. Y.S ran the simulations, conducted the post-processing analyses and wrote the first draft of the paper. E.G, J-B.S and J.M supervised the work and guided the choices made during the setup of the experimental design and its implementation, with the help of G.M. All authors contributed to the paper in its final form.

*Competing interests.* The authors declare that they have no conflict of interest.

*Acknowledgements.* Y.S. wishes to thank Alex Todd, Oleg Saenko and Duo Wang for their help with the experimental design and for answering questions about the fixed-flux forcing of the ocean model. We thank Matthew Couldrey for providing the FAFMIP multi-model mean data. We also thank Casimir de Lavergne and Jan Zika for helpful scientific ideas and discussions. This work was granted access to the HPC resources of IDRIS under the allocation 2020-A009017403 and 2020-A0080107451 made by GENCI. This work also benefited from the ESPRI computing and data centre (https://mesocentre.ipsl.fr) which is supported by CNRS, Sorbonne University, Ecole Polytechnique and CNES and through national and international grants. We acknowledge funding from the ARCHANGE project of the "Make our planet great again" program (ANR-18-MPGA-0001, France) as well as from the European Union's Horizon 2020 research and innovation program under grant agreement no. 821001.

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
