# Peer review of "A modeling framework to understand historical and projected ocean climate change in large coupled ensembles"

_EGUsphere, 2022_

## Author Response (AR1)

**Response to reviewers**

"A modeling framework to understand historical and projected ocean climate change in large coupled ensembles"

Yona Silvy, Clément Rousset, Eric Guilyardi, Jean-Baptiste Sallée, Juliette Mignot, Christian Ethé and Gurvan Madec

In this document, reviewer comments have been copied in black, author responses are in blue and citations from the revised manuscript are in green.

We thank the three reviewers for their comments and suggestions. We address two recurring comments of the reviewers here so as to cite the text and new figures once, before addressing each specific comment in the remainder of the document.

One of these comments was about the clarification of the scientific questions and goals of this work, and particularly the clarification of the investigation of the hist+ssp period. We have addressed these issues by a full reorganization of Section 2 of the manuscript. We have made an effort to better present the goals and general protocol in the general context of coupled models, before applying this protocol to IPSL-CM6A-LR and its specificities. We have also clarified the abstract, introduction and discussion sections when sentences were too vague.

We copy here the new section 2.1 presenting the experimental design and hope this clarifies the goals of our study.

[revised manuscript text omitted]

Another recurring comment regarded the validation of the simulations and the passive tracers, and their comparison to FAFMIP. We have now added a full comparison of the sensitivity simulations and of the passive temperature tracer with FAFMIP in the corresponding sections, adding two figures showing side by side our results compared to the FAFMIP multi-model mean of Couldrey et al. (2021). We have also added a figure to show the similarity between ALL and HEAT+WATER+STRESS in SST and SSS anomaly over time, to illustrate the linear additivity at each time step. We copy here these new additions to the manuscript.

L526-562:
"In terms of scientific validation, we verified whether:

1. the sensitivity experiments were phased in terms of internal variability, as explained in the goals of the study in section 2, so as to extract the estimate of the externally-forced component by taking the difference from CTL;

2. the linear-additivity hypothesis was correct, i.e. is the response to all the forcings (ALL) equal to the sum of the responses to individual forcings (HEAT+WATER+STRESS or BUOY+STRESS);

3. our results were comparable to other similar studies in the scientific literature in terms of patterns of long-term change.

To address point 1, an example is shown in Fig. 10b for SST, in which we can see that ALL, HEAT, WATER and STRESS are well phased with CTL in terms of interannual variability when ignoring their respective long-term trends caused by the climate change perturbations. This is also the case for the other variables, which we do not show here for readability.

Point 2 is partly addressed in Fig. A1 of Silvy et al. (2022), showing that for temperature, differences between ALL and HEAT+WATER+STRESS are mostly significant towards the end of the simulations in polar regions, where strongly non-linear interactions take place. By comparing [HEAT+WATER+STRESS - ALL], [BUOY+STRESS - ALL] and [HEAT+WATER - BUOY], we can trace which flux interactions cause these non-linearities. For example, we show that non-linearities in the subpolar North Atlantic are created by the interaction of heat and freshwater flux perturbations, while wind stress perturbation plays a minor role; on the contrary, non-linearities in the subpolar Southern Ocean arise from the interaction of all three surface flux perturbations. We add time series of SST and SSS anomaly (Fig. 13), comparing ALL (brown) with the sum HEAT+WATER+STRESS (dotted black). These two curves remain close together during the entire period, and differences, suggesting non-linearities, are much smaller than the intermember spread (orange shading). This demonstrates that the decomposition of processes in ALL using these sensitivity simulations can be made confidently. As shown in Silvy et al. (2022), regions of deep convection are the least additive. This indicates that the attribution of ocean changes to individual surface perturbations cannot be interpreted quantitatively in these regions as accurately as in other regions. Nonetheless, this mechanistic decomposition is still a useful tool to understand the physical causes of ocean regions.

[Figure]

Figure 13: Evolution of global mean sea surface temperature (a) and salinity (b) anomalies, excluding the Arctic, for the ocean-only ALL simulation (brown), HEAT (red), WATER (blue), STRESS (grey), the sum of HEAT, WATER and STRESS (dotted black), and the coupled model historical ensemble mean (orange). The ensemble mean anomaly is taken as the difference at each time step from the 1850-1899 average. The anomalies in the ocean-only experiments are computed at each time step as the difference between each experiment and CTL, before removing the 1850-1899 average, to be consistent with the ensemble mean anomaly.

Finally, for point 3 we compared the long-term response of these sensitivity experiments to what was done previously in the scientific literature in similar numerical designs (e.g. Fyfe et al. 2007, Armour et al. 2016, Gregory et al. 2016, Liu et al. 2018, Shi et al. 2020, Todd et al. 2020). Note however that most studies focused on heat storage and not on salinity. Here, we show the long-term change in ocean heat content in HEAT, WATER and STRESS in Fig. 14a,c,e (also see Silvy et al. 2022 Fig. 4 for zonal mean temperature). Although the amplitude, timescale and implementation of the forcings are very different, these results are very coherent qualitatively to previous studies cited above in terms of spatial patterns. The common patterns are described in more details in Silvy et al. (2022). We show furthermore the similarity of our results to the FAFMIP multi-model mean of Couldrey et al. (2021) in Fig. 14. While keeping in mind that the amplitudes of ocean heat content change cannot be directly comparable between the two studies (different forcings, timescales and configurations), and that we are looking at a single model versus a multi-model mean results, the patterns of change in HEAT, WATER and STRESS very closely match those of the FAFMIP results. Differences are expected from varying details in the protocols, but the similarity with FAFMIP and with other studies at the first order scientifically validates the long-term response of these simulations. This enables us to investigate the mechanistic decomposition of the historical+scenario response to anthropogenic climate change and its time evolution which were left unexplored in past modelling designs."

[Figure]

Figure 14: Vertically-integrated ocean heat content anomalies in the HEAT, WATER and STRESS experiments in this study (a,c,e) and in the FAFMIP multi-model mean of Couldrey et al. 2021 (b,d,f, which correspond to the data in their Fig. 6a-c). The mean is performed over 10-12 global climate models (see details in Couldrey et al. 2021). Anomalies are averaged over the period 2001-2100 for this study, and years 61-70 for the FAFMIP simulations. In a,c,e, stipples indicate where the anomaly is lower than twice the interannual standard deviation of the CTL experiment.

Regarding the tracers:

L613-629:

"[...]

Thus we can decompose the total temperature change T' in ALL, HEAT or BUOY as $T' = T'_{a} + T'_{r}$ (Total = Added + Redistributed). We can further decompose the added component $T'_{a}$ into an added heat component in the absence of circulation changes ($T'_{a}|_{CTL}$ or "passive heat" to be coherent with the FAFMIP terminology), and the remaining added heat due to the reorganization of added heat in response to circulation changes ($T'_{a}$ - $T'_{a}|_{CTL}$, referred to as "non-linear added heat" since it is a 2nd-order term). We can then write $T' =

$(T'_{a}|_{CTL}) + (T'_{a}$ - $T'_{a}|_{CTL}) + T'_{r}$, or "Total = Passive + Non-linear added + Redistributed".

Technical validation of PAT is not straightforward since there is no reference to compare it quantitatively in the model. One obvious check is to see whether the integral of PAT is equal to that of T' (Eq. 8), which is verified (not shown). We scientifically validate our PAT tracer by comparing its long-term spatial distribution with previous studies, even though they mostly focused on the response to idealized forcings (e.g. Banks and Gregory 2006, Winton et al. 2013, Marshalls et al. 2015, Garuba and Klinger 2016, Gregory et al. 2016, Dias et al. 2020, Todd et al. 2020). We find very similar patterns in most regions (see Silvy et al. 2022 for a more complete description of the common patterns). For instance, the spatial patterns of the different contributions of ocean heat content change in the HEAT experiment (Total, Passive, Non-linear added and Redistributed) match fairly well the FAFMIP multi-model mean of Couldrey et al. (2021) (Fig. 15) although one could expect large differences due to the different forcings and models used in the two studies. This illustrates that the passive temperature tracer behaves as it should in this framework."

[Figure]

Figure 15: Decomposition of total, passive, redistributed and non-linear added heat components (see text for definitions). Shown are the components for vertically-integrated ocean heat content anomalies in the HEAT experiment in this study (a,c,e) and in the FAFMIP multi-model mean of Couldrey et al. 2021 (b,d,f, which correspond to the data in their Fig. 12a,c,e,g). Anomalies are averaged over the period 2001-2100 for this study, and years 61-70 for the FAFMIP simulations. In a,c,e, stipples indicate where the anomaly is lower than twice the interannual standard deviation of the CTL experiment.

We have also added a figure for passive salinity (Fig. 16).

L653-656:

"There is no scientific literature to compare our passive salinity tracer to. In Fig. 16 we show its patterns in CTL (e.g. "Passive" salinity change $S'_{a}|_{CTL}$), which clearly illustrate in the upper ocean the fingerprint of water-cycle amplification in response to climate change with dry/salty regions getting saltier (subtropics) and wet/fresh regions getting fresher (equatorial and polar regions), as described for example in Eyring et al. (2021)."

[Figure]

Figure 16: Vertically-integrated (a) and zonal mean (b) passive anomaly salinity in years 2040-2059 of the CTL simulations. Stipples indicate where the value is lower than twice the interannual standard deviation of the salinity in the CTL experiment.
* * *
**RC1**

General comments:

The manuscript descibes and validates a modeling framework for investigating transient ocean climate change in the IPSL model. In general, the paper does a good job describing the model. However, some parts of the validation (particularly the Passive Tracers part, section 5) need more validation. It would also be helpful to the reader if the authors included more details on the scientific goals of this model. This seems to be mainly included in a separate, in review article. However, without some idea of the scientific goals, it is difficult to assess if the model set up can achieve those goals. For example, the authors do not include interactive sea ice, and yet they are investigating the timing of trends in temperature and salinity. I would expect that melting sea ice impacts T and S trends, at least in some regions. The authors need to better justify why sea ice would not impact whatever question they are investigating. The formatting and writing of the manuscript are clear. With revisions, the manuscript could be publishable.

Specific comments:

Line 65: The authors do not use coupled sea ice in the model. Perhaps this is ok, depending on the scientific question, but it is difficult to evalulate as they never state a scientific question. More details on the intended use of this model is would be helpful.

To answer this specific comment but also part of the general comment, we would like to make clear that we indeed do not use an interactive sea-ice model in the ocean-only experiments (as in the ocean-only FAFMIP protocol) but, the ocean model is forced at the liquid interface with heat, freshwater and stress fluxes from the coupled model, i.e. including fluxes between the sea-ice model and the ocean model. These fluxes are diagnosed in the AOGCM (here IPSL-CM6A-LR) and prescribed to the ocean-only model. Just like fluxes between the atmosphere and the ocean in ice-free areas. This means that sea-ice melting trends are indeed seen by the ocean model in the ocean-only simulations, but there is no feedback of the ocean on the fluxes at the base of sea-ice. In a certain way, we focus on oceanic processes and consider all exchanges with the other components of the Earth

system as prescribed boundary conditions. This is now clarified in several parts of the manuscript (see below), including in particular the new section 2.1.

We would also like to clarify that we are not proposing a model but rather a protocol. We hope that the new section 2.1 clarified this point.

L106-112:
"The piControl fluxes are thus imposed at high enough frequency so as to reproduce the internal variability as accurately as possible (here we chose 3-hourly, i.e. twice the coupling frequency) at the liquid ocean interface (below the atmosphere and the sea-ice), during 251 years. Since the fluxes are extracted below sea-ice and imposed on the ocean, the sea-ice model component is excluded from the oceanic configuration. Note however that the imposed fluxes are extracted from a fully coupled AOGCM which itself includes an interactive sea-ice module, so that the ocean-only component we run is impacted by both atmosphere and sea-ice fluxes. This CTL experiment is initialized similarly as its parent piControl simulation."

L191-193:
"The ocean configuration set up for this study is based on the one used in IPSL-CM6A-LR, where we replaced the sea-ice component, as explained above, by ice-ocean fluxes extracted from the coupled model and prescribed to the ocean-only experiment) [...] ."

L306-308:
"As indicated in section 2, the interactive sea-ice model was replaced by prescribed fluxes in the forced configuration. This choice was made to ensure that the ocean sees exactly the fluxes exchanged with sea-ice in the coupled model. This is consistent with our main modelling goal which is to reproduce the internal variability of the coupled model in the forced configuration. "

Line 70: Is the 2000 years the spin up, or is it run for 2000 years after spin up?

This 2000-year piControl simulation is run after the spin-up, consistently with the CMIP6 protocol. We have made that clearer in the text.

L166-168:
"For the purpose of the CMIP6 exercise, after a long spin-up, multiple experiments

were already conducted with this coupled model (Boucher et al. 2020) including: 2000 years of piControl simulation, [...]".

Line 71: what is ssp245? Perhaps include a citation

ssp245 is a specific scenario of future climate change. We have added the Gidden et al. (2019) citation.

Line 72: "isolate the mechanims responsible for temperature and salinity TRENDS"? Not sure what this sentence means without "trends" added to it

We have rephrased as follows:

L65-66:
"Our goal here is to investigate and isolate the mechanisms responsible for ocean temperature and salinity changes and their emergence from internal variability in historical+scenario simulations of single climate models."

Line 77-79: Why is it crucial to compare simulations with the same background internal variability? Perhaps it is important for what the authors are investigating, but I do not know what they are investigating so this statement confused me. It seems to me that it would be much more realistic to couple the ocean to the atmosphere. Please provide more justification for why this scientific question cannot be done with a coupled atmosphere.

It is indeed more realistic to couple the ocean to the atmosphere, which is done in the coupled version of the model. However, as now explained in the text, the attribution of the various fluxes could not be done in a coupled model because of varying feedback. Here, we describe a separate set-up designed to better understand the changes seen in the fully coupled model, without any retroaction to the forcings of the historical+scenario simulations. This point is clarified in the new section 2.1 (see citation in the beginning of the document).

Line 90: Why monthly-mean anomalies? Again, this may be appropriate for the scientific question, but I don't know what the scientific question is.

The monthly frequency was chosen as this is the highest-available frequency in the needed CMIP6 outputs. This choice seems reasonable since these monthly-averaged anomalies are meant to extract the slowly-varying externally-forced response only, not the high-frequency variability as is needed for

the CTL/piControl fluxes. Monthly averages are also commonly used to apply perturbations in similar modelling studies (e.g. Gregory et al. 2016, Fyfe et al. 2007, Xie and Vallis 2012, Garuba and Klinger 2018).

Line 96-98: Are you assuming that HEAT, STRESS, and WATER are all linear?

These experiments in fact cannot be perfectly additive, because the climate system itself is not linear. We address this point in section 4.3. We have clarified this in the text.

L129-131:
"Under the hypothesis of linear additivity, the sensitivity experiments decompose the response of the ALL experiment, while keeping identical phases of internal variability. This hypothesis and its limits are addressed in section 4.3."

Line 141: Is this slight difference in a figure somewhere?

It is shown in Table 2, as the difference in total incoming freshwater/heat/salt fluxes between piControl and CTL, and pointed out again e.g. at the end of the freshwater fluxes section: "Furthermore, piControl and CTL have quasi-identical net incoming freshwater fluxes (1.148 mSv vs 1.144 mSv respectively), the difference being due to the global ocean area which includes closed seas in the coupled configuration while it does not in the stand-alone ocean configuration, as stated before."

We have clarified the sentence pointed out by the reviewer as follows:

L202-203:
"A slight difference in total incoming heat, freshwater and salt fluxes is however detected (as shown in Table 2) and is solely due to a slightly larger global ocean area in the coupled configuration which includes closed seas."

Figure 2: There are many acronyms on this that I cannot find defined in the paper (e.g. ORCHIDEE, PISCES, LIM3, LMDz, OPA). Please define in the figure caption, or somewhere else in the manuscript.

We have defined these acronyms in the figure caption as requested.

Figure 2: Is the Chlorophyll forcing used? I thought that biogeochemistry was not included in the model version used (e.g. line 65)

Figure 2 is a schematic of the model components constituting the IPSL-CM6A-LR coupled model, as well as the exchanges between the ocean physics and the other components. As stated line 159, the biogeochemistry is indeed included in the coupled model. In the ocean-only simulations, we do not run the biogeochemistry but we prescribe the chlorophyll from the coupled model as represented by the green arrow in Fig. 2. As described in more details in section 3.5.2, the chlorophyll field is needed to compute the penetration of solar heat flux into the ocean.

Line 169: Does use of monthly means make a difference for ALL?

Yes, just like for CTL, using monthly means instead of daily means introduces errors in the evaluation of the difference in global quantities between the end and the beginning of the simulation. For ALL, we obtain an equivalent volume change ΔV of 5.417 mSv with daily means, vs. 5.244 mSv using monthly means, which results in an error of 0.174 mSv, similar to the 0.154 mSv error found for CTL.

Line 251: What are the "scientific questions we aimed to answer" ? I would expect that the timing of things like warming of the ocean may depend on having interactive sea ice. Please provide justification that no sea ice is needed for the scientific questions.

This protocol is designed precisely to mimic as closely as possible the time evolution of a fully coupled model (including interactive sea ice of course). We have now summarised the conclusions of the Appendix to clarify our choice here.

L317-322:
"The unconstrained case presented here turned out to better reproduce the coupled model than the alternative methods, both in amplitude and timing of key diagnostics, for the CTL and ALL simulations, apart from the surface Arctic Ocean. Furthermore, the unconstrained case allows a cleaner comparison with the passive temperature tracer. Consequently, this method is better suited to investigate the timing of physical changes in the ocean interior, their emergence from background climate variability and their attribution to different forcings, as compared to the other methods presented in Appendix A."

The sea ice issue has also been clarified in several instances, as explained above.

Line 310: What is the "purpose of the intended study"?

We have clarified the scientific questions and the main goals of this study in Section 2.1 (this entire section has been copied at the beginning of this document).

Figure 7: Differences appear very large in some areas (mostly Arctic, as well as high latitude N Atlantic and some of the Pacific). It is difficult to assess if this matters since I do not know exactly what the authors intend to investigate as their scientific question.

We have rephrased the end of this section as follows.

L372-383:
"The differences in zonal mean temperature and salinity (Fig. 7b,d) between the CTL and the reference piControl confirm that the largest errors ($\sim$0.2-0.5 ºC and $\sim$0.1 g/kg difference) are located in the Arctic and subpolar regions. The vertical dipolar structures suggest water-mass re-adjustment. There is initially some propagation of these differences at depth especially in the deep convection zone between 60-70 ºN, but the differences don't increase in time after they are installed (not shown). This also confirms that the CTL stays very close to the piControl in all other parts of the ocean, with very small differences between the two experiments (<0.05 ºC and <0.01 g/kg outside the surface subtropical gyres). For temperature, these differences are generally smaller than the piControl interannual variability. Localized differences in the Northern high latitudes exceed this threshold because of strongly non-linear dynamics, both for temperature and salinity. For salinity, differences also exceed this threshold in the global ocean interior, even though the amplitude remains quite small in absolute values. The reason why temperature seems to be better than salinity at reproducing the coupled model in the global interior remains difficult to be explained. Most importantly, these differences overall do not expand in time, and the CTL climate is taken as the new reference for the other ocean-only experiments."

Line 423-425: What is this work designed to study?

In this sentence, we mean to say that because the trends in the surface Arctic ocean are not well reproduced, this region should be kept out of any analysis. As for the general scientific questions, as mentioned above, we hope that the reviewer is now convinced that they have now been clarified, in particular through the new section 2.1.

Line 434-435: Are there citations of these previous experiments?

We have added the references to these studies, and have added a comparison with the FAFMIP multi-model mean in this section (see general response in the beginning of the document).

Section 5 ("passive tracers"): This section is intended to validate the passive tracers. However, there are no figures to illustrate that the tracers work as intended. Recommend adding at least one figure to illustrate that the passive tracers work as they are meant to.

We have now modified this section and added a comparison with the FAFMIP multi-model mean response (see general response in the beginning of the document).

Line 553-554: You have different scientific questions that FAFMIP. What are your scientific questions? What are those of FAFMIP? How are they different?

The scientific questions and their difference from those of FAFMIP have been further clarified in Section 2.1 (see citation from this new section in the beginning of the document), as well as in the introduction. We have made the summary of the main differences with FAFMIP in the discussion clearer (see next comment).

Line 561: "This question of timescales is precisely the novel aspect we aim to tackle with this present study": timescales of what? Will sea ice melting potentially impact this timescale? What region are you planning to look at?

We have clarified this sentence and reorganized this paragraph. We now hope that it is clear from the manuscript that we are interested here in emergence timescales. Regarding the question of sea-ice, we have answered in the previous comments by clarifying that melting terms are included in these simulations.

L700-705:
"Investigating the evolution in time of the balance of mechanisms causing ocean changes is precisely the novel aspect we have tackled with the present study. Additionally, we propose a framework to explore the mechanistic attribution of ocean changes as effectively seen in individual coupled models during their simulations of the historical period and future projections. This entails implementing exactly the perturbed fluxes as they are seen in the coupled model, as opposed to the generic FAFMIP perturbations. The results obtained from our experiments can be broadly compared with those of FAFMIP simulations in the

long-term response, and contain additional information on the transient response during the 1850-2100 period."

I selected "poor" for reproducibility as the manuscript currently does not comply with the journal's "Code and data availability" policy.

We have now made more efforts to comply with the Code and data availability policy.
* * *
**RC2**

In this manuscript, a modeling framework is designed to investigate the transient ocean climate change and the effects of different surface flux perturbations on it. Validations of designed experiments but not all are provided based on the results from the IPSL model. Overall, the manuscript is well organized in format and the writing is clear. The idea about the application of the hist+ssp245 scenario to the ocean-only model is also interesting here. However, I think the scientific goals are still vague, as also mentioned by another reviewer, especially when I compare this modeling framework with the FAFMIP&Ocean-only FAFMIP experiments. The interactive sea ice could be an issue here, but since another reviewer has raised several questions about that, I mainly provided some comments on the treatment and interpretation of internal variability and externally forced responses. Revisions are necessary to make the manuscript publishable.

Major Comments:

1.   I think the key to making this modeling framework different from the ocean-only FAFMIP is the usage of hist+ssp245. However, the manuscript took me a while to confirm the guess when I read it. It would help readers understand this work better if the authors can mention this difference in the abstract and at the beginning of the manuscript.

We apologize if this essential information was not made clear enough. We have made some important changes in the manuscript to clarify this point. In particular, we have clarified the abstract, the introduction, added a section 2.1 (see beginning of document) to present in simple terms the protocol, and clarified the discussion where we summarize the differences with FAFMIP.

e.g. some extracts of the changes:

L4-7 (abstract)
"While it has been partly addressed by modeling studies using idealized CO2 forcings, the time evolution of these individual contributions during historical and projected climate change is however lacking. Here, we propose a novel modeling framework to isolate these contributions in coupled climate models for which large ensembles of historical and scenario simulations are available."

L42-45 (introduction)

"One important goal of FAFMIP is thus to investigate the uncertainty in ocean responses across models to a unique set of perturbations. Strictly speaking, these perturbations are thus not coherent with the climate of each individual climate model, and the FAFMIP protocol does not enable to attribute the mechanisms responsible for ocean change within single coupled climate models. Furthermore, the FAFMIP protocol does not focus on the historical period but on an idealized forcing."

L701-705 (discussion)

"Additionally, we propose a framework to explore the mechanistic attribution of ocean changes as effectively seen in individual coupled models during their simulations of the historical period and future projections. This entails implementing exactly the perturbed fluxes as they are seen in the coupled model, as opposed to the generic FAFMIP perturbations. The results obtained from our experiments can be broadly compared with those of FAFMIP simulations in the long-term response, and contain additional information on the transient response during the 1850-2100 period."

2.    In the abstract, you mentioned "The question of timescales … is lacking" and may imply that these designed experiments are helpful to solve this. Is it a scientific goal here? However, I didn't find any related investigation or discussions in the main body. I believe that the validations under different timescales are needed if it is the scientific goal.

We agree we were unclear when referring to timescales. Indeed, investigating emergence timescales in the ocean is the scientific goal of our study. We hope that the reviewer will agree that this point has now been clarified in the introduction, section 2.1 and conclusive sections.

Concerning the validations at different timescales, we would like to stress that maximum divergence of the ocean-only framework from the coupled model response that we are trying to reproduce because of unperfect forcings and non-linearities is expected at the end of the simulations. Hence, the ALL experiment is validated (in the sense of conservation) in that we show the differences at the "worst" possible time. See also our response to comment #4 with a figure showing that differences are smaller earlier in time. We have changed this formulation where we were using it to avoid confusion. In Figure 10, we

showed time series of several diagnostics illustrating that the ALL simulation was well within the spread of the large ensemble. We have now added on these plots one individual member of the large ensemble to show that the behaviour of ALL was comparable to that of individual members. We have also added Figure 13 showing time series of global mean SST and SSS anomalies to compare the evolution of the forced response of the ocean-only framework vs. the forced response of the coupled model (see response to comment #4).

3.    I agree that the usage of the large ensemble can help obtain the forced responses. However, the multi-model mean variables used in FAFMIP also do the same thing. Are there other aspects to show the advantage of using large ensemble simulations in this study?

We agree that there can be several ways to estimate forced responses. By computing a multi-model average as in FAFMIP, we obtain a generic estimate of the forced response which is however not specific to the forced response of a particular model. In the framework proposed in this study, we wish to decompose the ocean response to different forcings effectively seen in a specific coupled model during the historical+scenario period. Hence, to construct the perturbation fluxes, we extract the forced response of this single coupled model during this historical+scenario period, so that the prescribed flux perturbations are coherent with the model in question. The goal of FAFMIP is different, they want to explore the uncertainties in ocean responses (as simulated by different models) to a unique set of idealized perturbations. They thus chose to construct these perturbations with a multi-model mean (from 1%CO2 experiments at the time of atmospheric CO2 doubling), to smooth out the forcings from different climate models and apply identical perturbations to different models. In doing so, they do not explore the ocean response to the perturbations in a way that is coherent with each coupled model's climate.

4.    Some figures about perturbation experiments show the anomalies over the period when the CO2 forcing is quite large, for example, the 2040-2059 average in Fig. 11. When the external forcing is strong, isolating forced response from internal variability is easier. Hence, I am wondering if the differences between ALL and IPSL ensemble mean are still small and insignificant when you look at the period when CO2 forcing is not that strong, such as around the mid-20th century. In addition, how will it become when you only look at the difference in one year, not the average of 20 years or so.

In other words, can this only one realization in ALL (HEAT, STRESS...) capture the result from the ensemble mean of 32 or 11 members at all time steps? Some validations are needed.

We chose to show validation for the mid-21st century because this is when differences between the forced response in ALL vs. the ensemble mean can start to sensibly diverge. Indeed, as there are no retroactions in the CTL nor in the ALL experiments, this divergence tends to increase in time. For example, we show below the same plots as in Figure 11, but for the anomalies in 1940-1959 as suggested (note the colorbar scale is 4 times smaller). We can see that the difference between the ocean-only setup and the coupled model forced response is this time smaller than the intermember spread everywhere in the ocean (stipples in the right column), apart from a negative temperature anomaly in the subsurface centered on 70°N which surpasses the spread.

[Figure]

The ocean-only setup thus captures the coupled model forced response better in the mid-20th century than later during the 21st century when the forcing is stronger. We have clarified our choice of validation period as follows:

L473-476:

"We show these diagnostics towards the end of the simulation (and at the end of the period over which the large ensemble is available) since it is when the ocean-only framework can most diverge from the coupled model response that we are trying to reproduce, as there are no retroactions in the CTL nor in the ALL experiments, which means differences tend to increase in time."

To address the second part of the comment, we also show below the anomaly in the year 2050 (instead of the 20-year average of Fig. 11). This compares extremely well to the 20-year average, with no larger difference between the forced response in ALL and IPSL-CM6.

[Figure]

The 20-year average was shown out of practice, since we rarely show anomalies in single years. However the reviewer is correct in asking about whether we can capture the coupled model response with the ocean-only simulations at all time steps. In regions where non-linearities are not too large, this is the case for most of the period considered, as the difference ALL-CTL theoretically removes the background variability prescribed in both simulations, and extracts the forced

response imposed in ALL only. This is why differences computed over a single year as the right column here hardly differ from the 20-yr average of Fig. 11 (only difference coming from the different period). However, as we explain in the manuscript, there are limitations to our methodology.

To clarify this point, we have:
- Added to Fig. 10b the sensitivity simulations, illustrating clearly that they all inherit the interannual variability from CTL, meaning that their difference from CTL extracts the externally-forced signal imposed from the perturbations.

[Figure]

- Added Fig. 13 (see below) showing the anomaly in SST and SSS, and comparing in particular ALL (brown line) to the ensemble mean (orange line).

[Figure]

Figure 13: Evolution of global mean sea surface temperature (a) and salinity (b) anomalies, excluding the Arctic, for the ocean-only ALL simulation (brown), HEAT (red), WATER (blue), STRESS (grey), the sum of HEAT, WATER and STRESS (dotted black), and the coupled model historical ensemble mean (orange). The ensemble mean anomaly is taken as the difference at each time step from the 1850-1899 average. The anomalies in the ocean-only experiments are computed at each

time step as the difference between each experiment and CTL, before removing the 1850-1899 average, to be consistent with the ensemble mean anomaly.

- Added some text related to this figure in the validation of the ALL experiment: L501-509:
"When averaging SST and SSS everywhere outside of the Arctic (Fig. 13), we can see that the forced response of the ALL experiment (brown line) reproduces well the forced response of the coupled model (orange line) at every time step. A small difference is seen in the 21st century for SST but does not grow much in time, and stays smaller than the intermember range. The general behaviour and time evolution of the coupled model is thus well reproduced by the ocean-only framework for SST outside the Arctic. For SSS, this difference is larger than SST compared to the total change and this remains to be investigated (see also section 3.6). These discrepancies might be very region-dependent depending on dynamical regimes more or less affected by the absence of ocean-atmosphere or ocean-ice feedbacks (e.g. the evolution in time of the coupled model forced temperature response is better reproduced in upper ocean mode waters than in deep and abyssal water-masses, Silvy et al. 2022)."

5.    How many ensemble members are sufficient to obtain the forced surface flux perturbation? Are 11 members also fine? In particular, there are only 11 realizations after 2060 but 32 before that. Could you show a map about the difference in perturbation between the 32-member-mean and 11-member-mean in the year 2059 (and 1959 when CO2 forcing is weak)?

Could you also discuss the effects of ensemble size on your results in ALL, etc.?

The question of the sufficient number of members needed to obtain the externally-forced response of a coupled model is a topic of research by itself (see e.g. Milinski et al. 2020). We would recommend that when applying the protocol described here, the maximum number of members available should be used and averaged to extract the best possible estimate of the externally-forced response to construct the perturbations (as explained in our new Section 2.1). In the case of IPSL-CM6A-LR, we took all the members at our disposal, i.e. 32 over the period 1850-2059. We showed in another work (Silvy, 2022) that 30 members were sufficient to reproduce the interannual variability of ocean interior temperature and salinity. We also chose to extend the simulations to 2100 with the 11 members available, to have the full 1850-2100 period completed, although as we have now clarified, this is not ideal and the focus of the studies using these experiments

should remain on the 1850-2059 period for more robustness. We leave the quantification of the sensitivity of the simulations to the number of members to potential future work, and for now advise to use as many as possible to construct the perturbations.

L170-176
"Thus, to apply the framework described previously, we have a large ensemble of 32 members over the period 1850-2059 to construct the surface flux perturbations. Since this is the period over which a large ensemble exists, we will mainly focus on this one. Because the regular ssp245 scenario was performed to 2100, but with 11 members only, we use the ensemble mean of these 11 members over 2060-2100 to complete our simulations to 2100. Using 11 members after 2060 is certainly less accurate than 32 members. Indeed, we have shown in previous work that the envelope of interannual variability of the piControl was well sampled with 30 members, and in particular that the piControl interannual standard deviation of ocean temperature and salinity is well reproduced by the intermember standard deviation (Silvy, 2022)".

6.   The results from HEAT, STRESS, and WATER are not shown. At least, the sum of them should be compared with ALL in a figure. If possible, HEAT, STRESS, and WATER may be comparable to the corresponding runs from ocean-only FAFMIP. For the passive tracers runs, the sum of added and redistributed terms should be compared with the HEAT or WATER in a figure, at least.

We refer to our complete response to this recurring comment in the beginning of the document.

Minor comments:

Title and Abstract: "large coupled ensembles" in the title is better to be also mentioned in the abstract.

We have added that in the abstract:

"Here, we propose a novel modeling framework to isolate these contributions in coupled climate models for which large ensembles of historical and scenario simulations are available."

Line 33: What do you mean by "decomposition of mechanisms"?

We have clarified this sentence as follows.

L35-36:
"A gap however remains regarding the precise attribution of the ocean response to these different forcings during the historical period and future projections."

Lines 35-36: "remain unclear" to "are not fully understood"? There are some findings from previous studies.

Thank you for the suggestion, which we have implemented.

Figure 5d,e,f: use dashed curve for CTL?

Thank you for the suggestion. We have homogenized figures 5 and 6, and used a dashed curve for the piControl.

Line 291: I feel the warming in the Pacific is not "slightly".

We removed "slightly". However, we note the difference is not significant compared to piControl interannual variability, as indicated by the stipples.

Line 294: "...than the piControl interannual variability" but in Figure 7 you use 2 x STDDEV.

We have clarified as follows.

L361-364:
"Nonetheless, in most parts of the world except for the warmer patch in the subpolar North Atlantic, the differences between the ocean-only CTL and the coupled piControl are smaller than twice the piControl interannual standard deviation and thus consistent with the distribution of interannual variability at the 95\% confidence level (represented by the stipples in Fig. 7)."

Line 366: "Differences may ...". Does this difference represent ALL-CTL minus ensemble mean anomaly or ALL-CTL?

The first one. We have made that clearer in the text.

L440-441:
"Differences between these two estimates of the forced response may however inevitably appear for several reasons."

Fig. 10a, c: Are these full-depth averages?

Yes. We have clarified this in the caption.

Line 408: "by twice the intermember standard deviation". I'm not sure whether the uncertainty range is too wide.

We use the same range as piControl variability (see earlier comment), i.e. two sigma which should encompass about 95% of the distribution.

Fig. 11: similar to major comment #4, how about the results after 2060?

After 2060 the 30 members for IPSL-CM6A-LR are not all available, so we have not analysed and dedrifted the coupled model outputs after this period. We prefered validating on the period where the large ensemble was available. The main object of this paper is a proof of concept of a modeling protocol; we leave more complete results to 2100 for other studies which would use a model where more scenario simulations are available.

Line 438: "This paper …" means this manuscript or the Silvy et al., in revision?

This is correct. This paper has now been published and is referenced adequately in the revised manuscript.
* * *
**RC3**

General comments:

This manuscript presents a modeling framework for investigating the effect of individual surface flux perturbations on transient ocean climate change, specifically focused on temperature and salinity trends. The modeling framework is implemented for the IPSL-CM6A-LR model such that the ocean component is forced with perturbations extracted from the historical+ssp245 ensemble of coupled model simulations. Overall, the structure and writing of the manuscript are clear, and most of the set-up is well validated. However, as discussed by previous reviewers, the scientific question that the new modeling framework seeks to address needs to be made more explicit. Choices for the framework set-up should be justified more clearly in the context of the scientific questions. Finally, the section on passive tracers should include validation plots, and passive anomaly salinity needs to be more thoroughly described. This manuscript could be publishable after revisions.

We thank the reviewer for these general comments. Please refer to the beginning of this document for general answers on the scientific questions at the basis of this work and some additional validation.

Specific comments:

Line 65: What is scientifically added by using fixed fluxes from sea ice rather than coupling the sea ice model? Is this the same justification for not coupling to the atmosphere - i.e. to prevent retroactions that may change the phase of the internal variability?

Regarding the sea-ice model, we refer to our first answer to reviewer #1, with clarification in the manuscript, e.g.:

L306-308:
"As indicated in section 2, the interactive sea-ice model was replaced by prescribed fluxes in the forced configuration. This choice was made to ensure the ocean sees

exactly the fluxes exchanged with sea-ice in the coupled model. This is consistent with our main modeling goal which is to reproduce the internal variability of the coupled model in the forced configuration. "

More generally, the general argument for our forced framework is (i) reproducing the exact same phasing of the internal variability in various sensitivity experiments where time-varying individual fluxes are imposed separately and (ii) the cost of computation time. This is now extensively clarified in new section 2.1.

Line 94: Include validation on going from 32 members to 11 members after 2060, such as a plot showing mean and variability of forcings. As reviewer 2 asked, how many members are necessary to obtain the forced surface flux perturbation?

We copy here our response to reviewer #2 's comment #5 about that topic, and modifications made in the manuscript.

The question of the sufficient number of members needed to obtain the externally-forced response of a coupled model is a topic of research by itself (see e.g. Milinski et al. 2020). We would recommend that when applying the protocol described here, the maximum number of members available should be used and averaged to extract the best possible estimate of the externally-forced response to construct the perturbations (as explained in our new Section 2.1). In the case of IPSL-CM6A-LR, we took all the members at our disposal, i.e. 32 over the period 1850-2059. We showed in another work (Silvy, 2022) that 30 members were sufficient to reproduce the interannual variability of ocean interior temperature and salinity. We also chose to extend the simulations to 2100 with the 11 members available, to have the full 1850-2100 period completed, although as we have now clarified, this is not ideal and the focus of the studies using these experiments should remain on the 1850-2059 period for more robustness. We leave the quantification of the sensitivity of the simulations to the number of members to potential future work, and for now advise to use as many as possible to construct the perturbations.

L170-176
"Thus, to apply the framework described previously, we have a large ensemble of 32 members over the period 1850-2059 to construct the surface flux perturbations. Since this is the period over which a large ensemble exists, we will mainly focus on this one. Because the regular ssp245 scenario was performed to 2100, but with 11 members only, we use the ensemble mean of these 11 members

over 2060-2100 to complete our simulations to 2100. Using 11 members after 2060 is certainly less accurate than 32 members. Indeed, we have shown in previous work that the envelope of interannual variability of the piControl was well sampled with 30 members, and in particular that the piControl interannual standard deviation of ocean temperature and salinity is well reproduced by the intermember standard deviation (Silvy, 2022)".

Section 3.5.2/Fig 4: Here, validation of the method for prescribing the chlorophyll field is by comparing global SST for each option. Does temperature in the vertical agree?

Indeed, temperature at depth is also validated. We have now added to Figure 4 a new panel b showing vertical profiles of globally-averaged temperature in the first year of simulation, for the different methods. As for the surface, options 1 and 3 give unsatisfying results in the subsurface, with large anomalies compared to the reference piControl, anomalies that change sign below 50m with some clear vertical redistribution. In contrast, options 2 and 4 remain close to the piControl.

[Figure]

Line 250: It would be helpful to have more details here on why the unconstrained case is better for the scientific questions.

We have clarified this part as follows.

L317-322:
"'The unconstrained case presented here turned out to better reproduce the coupled model than the alternative methods, both in amplitude and timing of key diagnostics, for the CTL and ALL simulations, apart from the surface Arctic Ocean. Furthermore, the unconstrained case allows a cleaner comparison with the passive temperature tracer. Consequently, this method is better suited to

investigate the timing of physical changes in the ocean interior, their emergence from background climate variability and their attribution to different forcings, as compared to the other methods presented in Appendix A."

Fig 7d shows that for most depths below 1000-2000m, the zonal mean salinity difference is larger than 2 times interannual standard deviation of piControl. More justification on why this still validates the CTL experiment for the intended use would be helpful.

Salinity at depth is indeed less well reproduced in the ocean-only CTL than temperature, something which is not fully understood yet. Looking at salinity in these regions will thus have to be done with caution. We note however that these differences remain small in amplitude, and that this discrepancy between salinity and temperature at depth disappears in the ALL simulation (cf Fig. 11).
We have modified the paragraph describing this figure to reflect this comment.

L379-383:
"For salinity, differences also exceed this threshold in the global ocean interior, even though the amplitude remains quite small in absolute values (<0.01 g/kg). The reason why temperature is better at reproducing the coupled model in the global interior than salinity remains difficult to be explained. Most importantly, these differences overall do not expand in time, and the CTL climate is taken as the new reference for the other ocean-only experiments."

Line 361: Some mismatch between ALL and the fully coupled ensemble could also be due to eliminating atmosphere-ocean feedbacks. If the goal for the ALL experiment is to simulate a climate consistent with the coupled model ensemble, is this a limitation of this framework?

On the contrary, we believe including atmosphere-ocean feedbacks would generate different fluxes seen by the ocean as compared to the coupled model. This would also result in decoupling the variability between the different ocean-only experiments. We have now better explained this point in our new Section 2.1 (see beginning of document for the full section). Of course there are limitations to our framework, due e.g. to the forcings being prescribed at lower frequency than the coupling frequency in the fully coupled model, and which are addressed in the manuscript in beginning of subsection 4.2, and in the discussion.

Fig 9c and 9d: Visually, the freshwater fluxes for this study appear to be stronger than the FAFMIP anomalies especially in the Arctic. Is this due to a stronger model response than the CMIP5 ensemble mean or due to other differences in experimental set-up?

The reason for this difference is still unclear. It could be due to either a stronger model response than the CMIP5 ensemble mean, but could also be due to the forcings used here. Indeed, our perturbations are constructed from the hist+ssp experiments (in a single model), vs. the FAFMIP perturbations are constructed from the 1%CO2 simulations, at the time CO2 concentration has doubled in the atmosphere. This means that the flux perturbations are a response to different forcings (CO2 only and at different timescales for FAFMIP vs. all anthropogenic forcings in this study).

We have clarified this in the text.

L427-429:
"We recall that the latter are constructed from a multi-model mean of 1%CO2 idealized experiments, thus we expect differences with the perturbations in the present study due to different external forcings, timescales and models."

Line 423-425: There are also some significant differences in salinity in the Southern Ocean between ALL and the coupled model (Fig 11f and l)

We have added some clarification in this sentence to reflect on these differences.

L512-513:
"Interestingly, the absence of a sea-ice model is found to be much less problematic in the Southern Ocean (although we do note some significant differences in salinity in the deeper parts, Fig. 11f,l)."

Line 428: This manuscript just shows results from ALL. It is stated that comparing HEAT+WATER+STRESS with ALL is in the companion paper, but it would be helpful to show it here.

We have indeed added a validation of the linearity assumption in the manuscript, we refer to our response to this recurring comment in the beginning of the document.

Note that the companion paper (Silvy et al. 2022), is now published with open-access (https://journals.ametsoc.org/view/journals/clim/aop/JCLI-D-22-0074.1/JCLI-D-22-0074.1.xml).

Line 436: Which studies were compared to?

We have added the references to these studies and have also added a comparison with the FAFMIP multi-model mean (see beginning of document for the full citation of the revised manuscript and the new figure).

Sec 5: Validation plots for passive tracers needed.

We have now added a full comparison with the FAFMIP multi-model mean response in Section 5.1 (see beginning of document for the full citation of the revised manuscript and the new figures).

Line 495 to 500: Equations equivalent to Equations 6 and 7 for salt and PAS would help for understanding the additional complexities here.

Line 505: Similar to for PAT, is there PAS in all experiments which experiences F'? If so, make this a bit clearer.

In response to these two comments, we have revised the text to make the implementation of PAS clearer. As we explain, the formulation of salinity is different than for temperature because the ocean volume locally varies in time (variable volume formulation), which means that the forcing term for PAS is only the salt flux anomaly, and the freshwater flux forcings are taken into account in another part of the code which modifies the volume of the grid cells and thus the salt concentration.

L631-651:
"Similarly to temperature, the Passive Anomaly Salinity (PAS or S'a) is forced by the freshwater and salt flux anomalies (the same as the prognostic salinity sees in the perturbed experiments), and can be implemented in all simulations (even CTL) as PAS does not affect the dynamics.

However, the implementation of PAS is more complex than PAT, since the sources and sinks of PAS can be separated unlike for PAT. In NEMO, in particular, when the local ocean volume freely evolves in time (variable volume formulation), the

surface boundary condition for ocean salinity is the salt flux (sfx) only. However, ocean salinity is also locally influenced by freshwater fluxes (emp, runoff, iceshelf) through a concentration/dilution effect. Furthermore, for every simulation, PAS must only be affected by the externally-induced anomalies (sfx', emp', runoffs', iceshelf') and not by the background fluxes from the piControl. In particular, we do not want PAS to be impacted by the background freshwater fluxes, present in the concentration/dilution effect. Moreover, the concentration/dilution effect differs between experiments since the freshwater flux perturbations are used to force only some of the experiments (ALL, WATER, BUOY), and not the others (CTL, HEAT, STRESS). Hence:

1. PAS is forced by sfx' in its trend

2. to compensate the background concentration/dilution effect on PAS, we remove the effect of the piControl fluxes (emp, runoffs, iceshelf, without the anomalies) from the PAS trend;

3. to obtain the same effects of freshwater flux perturbations on PAS between experiments, we add freshwater flux anomalies in the PAS trend for CTL, HEAT and STRESS.

PAS is initialized to the approximate ocean global mean salinity (34.7 g/kg), and not 0, for the formulation of the freshwater flux anomaly to be correct. This mean value is removed in all post-processing analyses to obtain an anomaly. All the forcing terms are applied to the passive tracer trend in the same way as for salinity: sfx' and emp' act on the top ocean level while runoff' and iceshelf' are distributed vertically.

Technical corrections:

Line 21: "associated with" rather than "associated to"

Line 125: "bears similarities" rather than "bares similarities"

Line 266: capitalize "deacon"

We have implemented these three corrections, thank you.

Line 293: says stipples represents less than piControl interannual variability in Fig 7 but the figure caption says 2 times interannual standard deviation

We have clarified as follows.

L361-364:
"Nonetheless, in most parts of the world except for the warmer patch in the subpolar North Atlantic, the differences between the ocean-only CTL and the coupled piControl are smaller than twice the piControl interannual standard deviation and thus consistent with the distribution of interannual variability at the 95\% confidence level (represented by the stipples in Fig. 7)."

Line 610: "worse" rather than "worst"

Line 628: "constraints" rather than "constrains"

Thank you for these technical corrections, which we have made.

---

## Author Response (AR2)

**Response to reviewers [Revision #2]**

"A modeling framework to understand historical and projected ocean climate change in large coupled ensembles"

Yona Silvy, Clément Rousset, Eric Guilyardi, Jean-Baptiste Sallée, Juliette Mignot, Christian Ethé and Gurvan Madec

In this document, reviewer comments have been copied in black, author responses are in blue and citations from the revised manuscript are in green.

We thank the two reviewers for their time and for their final suggestions.
* * *
**Report #1**

Does the historical forced IPSL model contain all forcings (e.g., greenhouse gases, aerosols, and volcanos)? If it does, it should be mentioned somewhere that aerosols could play a role in results over the historical period. Perhaps it could be included in the introductory paragraph on the differences between FAFMIP and this set up, since I don't think FAFMIP include aerosol forcing.

The IPSL historical simulations indeed include all forcings mentioned here, as per the CMIP6 protocol. We have made that clearer in the introduction as suggested.

"Furthermore, the FAFMIP protocol does not focus on the historical period but on an idealized $CO_2$ forcing, and does not include non-$CO_2$ anthropogenic forcing agents such as other greenhouse gases and aerosols, which can play an important role on historical climate change patterns (Wang et al. 2016)."
* * *
**Report #2**

The authors have carefully addressed my concern in this revision. I'm delighted to find that more clarifications and results about validation are added, which improve the manuscript. I fully support its publication in EGUsphere after some additional edits.

We are glad that the changes made in the first round of revisions met the reviewer's expectations, and thank them for these additional suggestions.

(The following line numbers are based on the PDF file with tracked changes.)

Comments:

1. Line 111: How about adding "fully-coupled" before AOGCM?

Thank you for the suggestion, we added it.

2. Lines 123-127: Does it mean this approach only works for the model with large-ensemble fully-coupled simulations? How good or bad it would be if only 3~5 (or even 1) fully-coupled realizations are available to use this modeling framework?

As we explained in our first response to reviewers, using a large ensemble is the most appropriate way to extract the externally-forced signal of a model. Averaging over less than 10 members would certainly degrade the estimate of this forced signal. The question of the number of members necessary to extract that externally-forced signal depends on the variable (mean or variance for example), it is a complex question and it is thus not the focus of this study. We show that with 30 members of this model, we can sample the interannual internal variability of the model, and thus obtain an estimate of the forced signal that averages through these variations. There are other ways to estimate the externally-forced signal from a single member, such as fitting a 4-th order polynomial, but this has been shown to be less precise than using the average over a large ensemble (Lehner et al. 2020).

3. Line 134: Are you referring to "ALL experiments" (those ocean-only runs) or just "all experiments"?

We are referring to all the ocean-only experiments. We have clarified this in the text.

4. Lines 145-146: Isn't climate drift the major factor here?

We do not think so, as climate drift is the same in CTL and in ALL, thus the difference ALL-CTL removes the drift.

5. Lines 160-162: This last sentence seems to be out of place.

This sentence is a follow-up to the previous one mentioning the extraction of the externally-forced signal. We reorganized the sentence to make it more comprehensible.

"Nevertheless, as described above, the experimental design could be applied to any coupled model and its ocean-only configuration, as long as the externally-forced historical and future response can be extracted. To extract this forced response, the historical+scenario large ensemble approach seems to be the most accurate way compared to e.g. fitting a 4th order polynomial to a single member (Lehner et al. (2020))."

6. Caption of Fig. 3: I believe the fluxes shown here from the piControl and CTL are identical. How about rephrasing it to "... from the piControl and used in CTL" to clarify?

Yes, thank you for the suggestion.

7. Caption of Fig. 10: It would be helpful to point out how is the inter-member spread defined.

We have added: "full intermember spread (minimum to maximum)"

8. Caption of Fig. 11/12 and Lines 491-510:
Fig. 11/12 and the associated statements are based on the difference between 2040-2059 and 1850-1899, which reflects the projected climate change mentioned in the title. I'm still wondering what the historical responses look like, such as 1980-2000 minus 1850-1899. Based on Fig. 10, the external forcing seems relatively weak during the historical period. Does the ALL runs behave well during this period?

We refer to our response to comment #4 of Reviewer #2 in the previous response to reviewers (pages 23-26). The ALL experiment in fact behaves better during the historical period compared to the 21$^{st}$ century since it is mostly the forced signal, applied at monthly frequency, which creates non-linear responses, and so the stronger the signal gets the larger the discrepancies become, over time, between ALL and the ensemble mean in strongly non-linear regions.

9. Lines 535-538 and Lines 742-749: I don't think the Southern Ocean is only affected by a little in the absence of the interactive sea-ice model. As I mentioned in #8, the Southern Ocean situation may be worse during the historical time. In the last sentence (Line 537), how about also pointing out the Southern Ocean besides "the Arctic Ocean"? Similarly, the discussion in the last paragraph (Lines 742-749) should also include the Southern Ocean.

As illustrated in Figure 7, 11 and 12, the differences between the coupled configuration and the forced configuration are much larger in the Arctic than everywhere else, including the Southern Ocean. The Arctic stands out, which is not the case of the Southern Ocean (in which the differences are the same order of magnitude than in the global ocean), which is why we mention that the Southern Ocean, compared to the Arctic, seems to be only little affected by the absence of a sea-ice model. As mentioned in the previous comment, the response is in fact better during the historical period than during the scenario period, due to weaker non-linearities.

10. Caption of Fig. 13: Is it the global mean SSS/SST or the average within 60˚S-60˚N?

It is the global average south of 60ºN, we have clarified this in the caption, thank you for pointing this out.

11. References: There are two Silvy 2022 in the references. One is unpublished. If possible, please use a,b to show which one you are referring to.

Silvy (2022) refers to a thesis, deposited online, and Silvy et al. (2022) refers to a published paper in Journal of Climate. We do not have a hand on the format since we used the Copernicus latex template.